# Stress-induced formation of cell wall-deficient cells in filamentous actinomycetes

Karina Ramijan[1], Eveline Ultee[1], Joost Willemse[1], Zheren Zhang[1], Joeri A. J. Wondergem [2], Anne van der Meij[1], Doris Heinrich[2,3], Ariane Briegel[1], Gilles P. van Wezel [1] & Dennis Claessen [1]

The cell wall is a shape-defining structure that envelopes almost all bacteria and protects them from environmental stresses. Bacteria can be forced to grow without a cell wall under certain conditions that interfere with cell wall synthesis, but the relevance of these wall-less cells (known as L-forms) is unclear. Here, we show that several species of filamentous actinomycetes have a natural ability to generate wall-deficient cells in response to hyper-osmotic stress, which we call S-cells. This wall-deficient state is transient, as S-cells are able to switch to the normal mycelial mode of growth. However, prolonged exposure of S-cells to hyperosmotic stress yields variants that are able to proliferate indefinitely without their cell wall, similarly to L-forms. We propose that formation of wall-deficient cells in actinomycetes may serve as an adaptation to osmotic stress.

[1] Molecular Biotechnology, Institute of Biology, Leiden University, P.O. Box 9505, 2300 RA Leiden, The Netherlands. [2] Biological and Soft Matter Physics, Huygens-Kamerlingh Onnes Laboratory, Leiden University, P.O. Box 9504, 2300 RA Leiden, The Netherlands. [3] Fraunhofer Institute for Silicate Research ISC, Neunerplatz 2, 97082 Würzburg, Germany. Correspondence and requests for materials should be addressed to D.C. (email: D.Claessen@biology.leidenuniv.nl)

All free-living bacteria are challenged by constant changes in their environment, and their survival depends on the ability to adapt to sudden exposure to stressful conditions. For instance, soil bacteria can encounter rapid osmotic fluctuations caused by rain, flooding or desiccation. Bacterial cells typically respond to osmotic changes by rapidly modulating the osmotic potential within the cell, either by importing or exporting ions and compatible solutes[1]. While these responses typically occur immediately after cells have been exposed to the changed environment, they are also able to tune the expression of metabolic pathways or critical enzymes[2].

How such osmotic changes affect cellular morphology is not well known. The cells' shape is largely dictated by the cell wall, which is a highly dynamic structure that acts as the main barrier that provides osmotic protection[3]. The synthesis of its major constituent, peptidoglycan (PG), involves the activity of large protein complexes that cooperatively build and incorporate new PG precursors into the growing glycan strands at the cell surface[4–7]. These strands are then cross-linked to form a single, giant sacculus that envelops the cell[8]. The sites for the incorporation of new PG is a major difference between the planktonic firmicutes that grow by extension of the lateral wall, and Actinobacteria, which grow via apical extension and thereby incorporating new PG at the cell poles[9,10].

Actinobacteria display a wide diversity of morphologies, including cocci (Rhodococcus), rods (Mycobacterium and Corynebacterium) and mycelia (Streptomyces and Kitasatospora), or even multiple shapes (Arthrobacter)[11,12]. Species belonging to these genera are able to change their morphology to adapt to extreme environments. For example, Rhodococcus species that are commonly found in arid environments are able to adapt to desiccation by modulating their lipid content and form short-fragmented cells[13]. Arthrobacter species also exhibit high resistance to desiccation and cold stresses. Upon hyperosmotic stress, these cells can modulate the synthesis of osmoprotectants and switch between rod-shaped and myceloid cells[12].

While the cell wall is considered an essential component of virtually all bacteria, most species can be manipulated under laboratory conditions to produce so-called L-forms that are able to propagate without their wall[14–17]. Typically, L-forms are generated by exposing walled bacteria to high levels of lysozyme combined with antibiotics that target cell wall synthesis in media containing high levels of osmolytes[18,19]. Stable L-forms that can propagate indefinitely without the cell wall require two mutations that fall in separate classes[18]. The first class of mutations leads to an increase in membrane synthesis, either directly by increasing fatty acid biosynthesis or indirectly by reducing cell wall synthesis[20]. The second class of mutations reduce oxidative damage caused by reactive oxygen species, which are detrimental to proliferation of L-forms[21]. Notably, proliferation of L-forms is independent of the FtsZ-based division machinery[15,22]. Instead, their proliferation can be explained solely by biophysical processes, in which an imbalance between the cell surface area to volume ratio leads to spontaneous blebbing and the subsequent generation of progeny cells[20]. Such a purely biophysical mechanism of L-form proliferation is not species-specific. This observation has led to the hypothesis that early life forms propagated in a similar fashion well before the cell wall had evolved[15,20,23]. Whether L-forms have functional relevance in modern bacteria, however, is unclear.

Here, we present evidence that filamentous actinobacteria have a natural ability to extrude cell wall-deficient (CWD) cells when exposed to high levels of osmolytes. These newly-identified cells, which we call S-cells, synthesize PG precursors and are able to switch to the canonical mycelial mode-of-growth. Remarkably, upon prolonged exposure to hyperosmotic stress conditions,

S-cells can acquire mutations that enable them to proliferate in the CWD state as L-forms. These results infer that the extrusion of S-cells and their transition into proliferating L-forms is a natural adaptation strategy in filamentous actinobacteria caused by prolonged exposure to osmotic stress.

## Results

**Hyperosmotic stress induces formation of wall-deficient cells.**
Recent work suggests that hyperosmotic stress conditions affects apical growth in streptomycetes[24]. Consistent with these observations, we noticed that growth was progressively disturbed in the filamentous actinomycete Kitasatospora viridifaciens, when high levels of sucrose were added to the medium (Fig. 1a). In liquid-grown cultures containing more than 0.5 M sucrose, initiation of growth was delayed by at least 5 h compared with media with low levels of sucrose. A similar retardation in growth was observed on solid medium supplemented with high levels of osmolytes, evident from the size decrease of colonies (Fig. 1b, c). On average, their size decreased from 12.8 mm$^2$ ($n = 278$) to 1.4 mm$^2$ ($n = 184$) after 7 days of growth. Notably, the high osmolarity also reduced the number of colony forming units (CFU) by 33%, from $9.3 \times 10^8$ CFU ml$^{-1}$ to $6.1 \times 10^8$ CFU ml$^{-1}$, as deduced by plating serial dilutions of spores in triplicate. In order to study the morphological changes accompanying this growth reduction, we stained the mycelium after 48 h of growth with the membrane dye FM5-95 and the DNA stain SYTO-9 (Fig. 1d, e). The high levels of osmolytes had a dramatic effect on mycelial morphology. The hyphae showed indentations along the cylindrical part of the leading hyphae, reminiscent of initiation of sporulation (see BF panel in Fig. 1e). In addition, the branching frequency increased by more than threefold in the presence of high levels of osmolytes (Supplementary Fig. 1a, h; Supplementary Table 1, 2; Student's T-test, P-value = 0.0010). Additionally, we noticed that these stressed hyphae contained an excess of membrane (compare FM5-95 panels in Fig. 1d, e). The proportion of the hyphae that were stained with FM5-95 increased from 10% to 21% in the presence of 0.64 M sucrose (Supplementary Fig. 1e, l; Supplementary Table 1, 2; Student's T-test, P-value < 0.0001). Simultaneously, the average surface area occupied by the nucleoid decreased from 2.59 μm$^2$ to 1.83 μm$^2$, indicative for condensation of DNA (Supplementary Fig. 1g, n; Supplementary Table 1, 2; Student's T-test, P-value = 0.0074). Strikingly, we observed large DNA-containing vesicles surrounding the mycelial networks (indicated by arrowheads in Fig. 1e). High levels of the osmolytes NaCl (0.6 M) and sorbitol (1 M) caused a comparable growth defect (Supplementary Fig. 2a) and also led to the formation of DNA-containing vesicles (Supplementary Fig. 2b–d). Notably, addition of high concentrations of salt (0.6 M NaCl) differently affected morphology and yielded mycelial particles that were small and very dense (Supplementary Fig. 3). K. viridifaciens was no longer able to grow when the NaCl concentration was increased to >0.6 M (not shown). The formation of DNA-containing vesicles in the presence of both ionic (NaCl) and non-ionic, organic osmolytes (sucrose and sorbitol) indicate that the hyphae form a previously uncharacterized cell type upon hyperosmotic stress, which we hereinafter will refer to as S-cells, for stress-induced cells.

To distinguish S-cells from other CWD variants of K. viridifaciens, we compared them to fresh protoplasts and L-form cells obtained after classical induction with high levels of lysozyme and penicillin (see Methods). Size measurements from 2D images revealed that S-cells had an average surface area of 20.73 μm$^2$ ($n = 213$) and were larger than protoplasts ($n = 514$) and L-forms ($n = 678$), which had an average surface area of 4.01 μm$^2$ and 7.06 μm$^2$, respectively (Table 1). Vancomycin-BODIPY

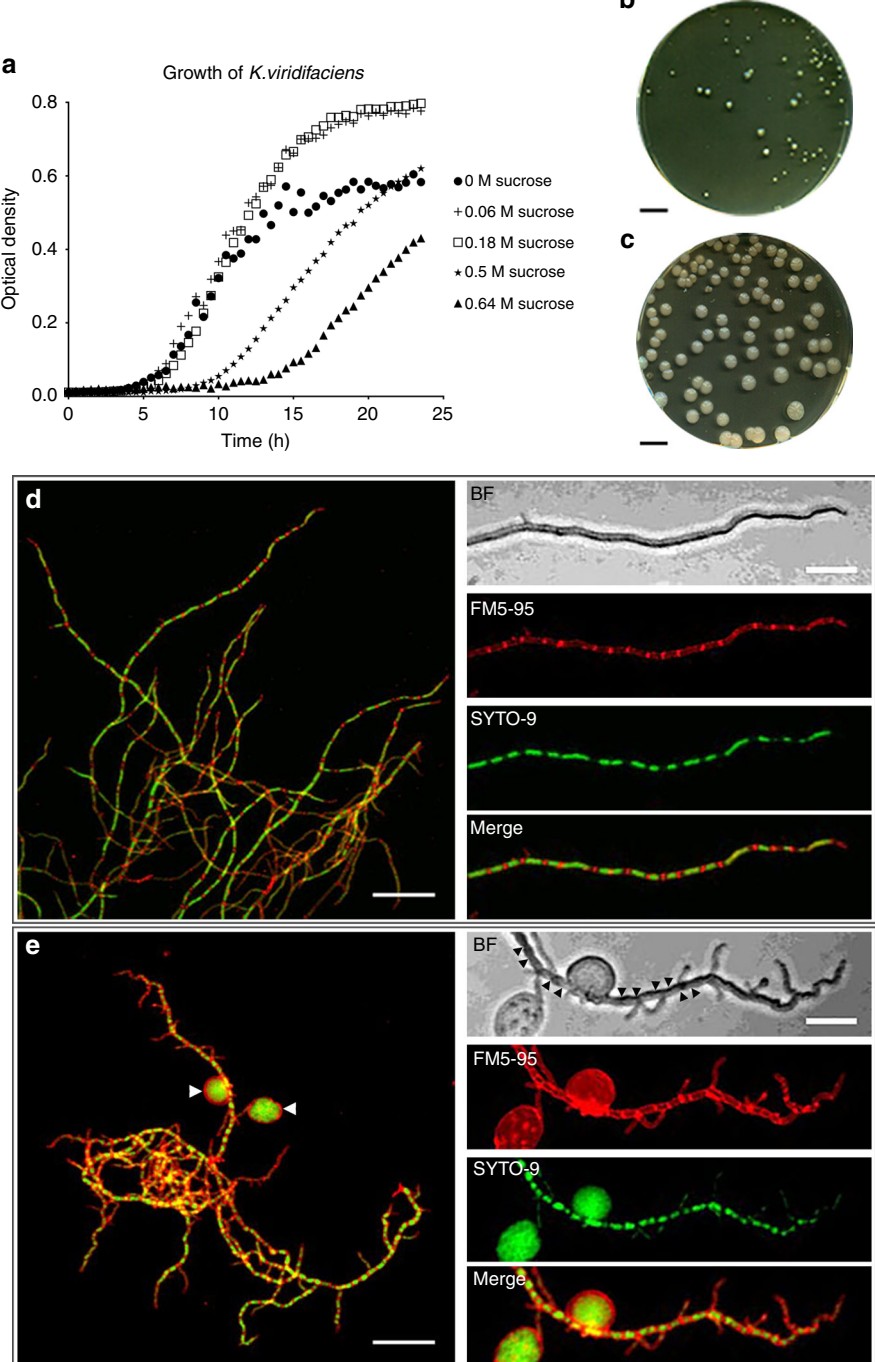

**Fig. 1** High levels of sucrose affect growth and morphology of *K. viridifaciens*. **a** Growth curves of *K. viridifaciens* in LPB medium supplemented with increasing amounts of sucrose. Values represent the average of five independent replicates. High levels of osmolytes reduce the number and size of colonies (**b**) in comparison to media without osmolytes (**c**). Mycelial morphology *K. viridifaciens* grown in LPB without sucrose (**d**) and with 0.64 M of sucrose (**e**). Mycelium was stained with FM5-95 and SYTO-9 to visualize membranes and DNA, respectively. Please note the S-cells (white arrowheads in **e**) and indentations along the cylindrical part of the hypha, indicated with black arrowheads in the brightfield (BF) section, formed in medium containing high levels of sucrose. Scale bars represent 10 mm (**b**, **c**), 20 μm (left panels in **d**, **e**) and 10 μm (magnified section in **d** and **e**)

staining (van$^{FL}$, Fig. 2a) revealed a heterogeneous pattern of nascent PG synthesis in these cells, while in L-forms mostly detached wall material was observed. By contrast, no staining was detected when freshly prepared protoplasts were used (Fig. 2a). When protoplasts were maintained in LPB for 48 h, their average surface area increased to 7.49 ± 2.21 μm$^2$, which is smaller than that of S-cells (Table 1). Furthermore, protoplasts regenerated a more uniform cell wall while S-cells showed a disordered, non-

uniform pattern of cell-wall assembly, whereby wall material was sometimes found to be detached from the cell surface (Fig. 2b, Table 1).

**Formation of S-cells is common in natural isolates.** To see how widespread the formation of S-cells is among natural isolates, we screened our collection of filamentous actinomycetes, obtained from the Himalaya and Qinling mountains[25], using *Streptomyces*

**Table 1 Comparison between *K. viridifaciens* cell wall-deficient cells**

| Characteristic | Protoplast | L-form | S-cell |
|---|---|---|---|
| Origin | Osmoprotective conditions combined with lysozyme treatment | Osmoprotective conditions combined with prolonged exposure to lysozyme and penicillin G | Osmoprotective conditions |
| Area ($\mu m^2$) | 4.01 ± 1.93 | 7.06 ± 5.87 | 20.73 ± 11.53 |
| Cell wall | Homogeneous regeneration. Wall material mostly associated with the cell surface | Not uniform, disordered assembly. Wall material often detached from the cell surface | Not uniform, disordered assembly. Wall material sometimes detached from the cell surface |
| Genotype | Wild type | Mutant | Wild type |

Bacteria can be forced to grow without cell wall if cell wall synthesis is inhibited. Here, Ramijan et al.[30] show that, in filamentous actinomycetes, hyperosmotic stress induces formation of wall-deficient cells that can switch to normal mycelial growth, or mutate and proliferate indefinitely as wall-less forms.

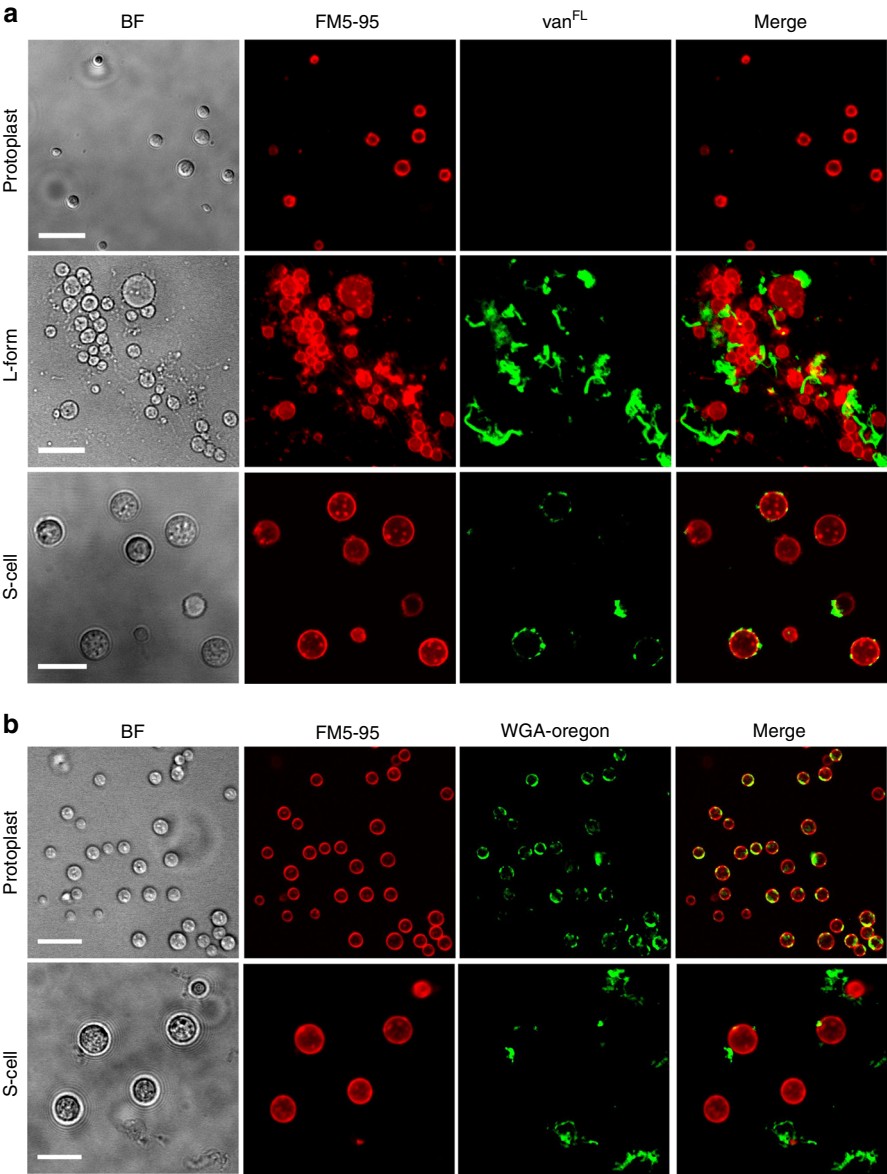

**Fig. 2** Comparison between different cell wall-deficient cell types in *K. viridifaciens*. **a** Morphology of freshly made protoplasts (top panels), penicillin-induced L-forms (middle panels) and S-cells (bottom panels). Cells were stained with the membrane dye FM5-95 or fluorescent vancomycin (van^FL) to detect nascent PG. **b** Morphology of protoplasts (top panels) and S-cells (bottom panels) grown for 48 h in LPB containing 0.6 M sucrose. Cells were stained with the membrane dye FM5-95 or wheat germ agglutinin (WGA-Oregon) to detect PG. Scale bars represents 10 μm

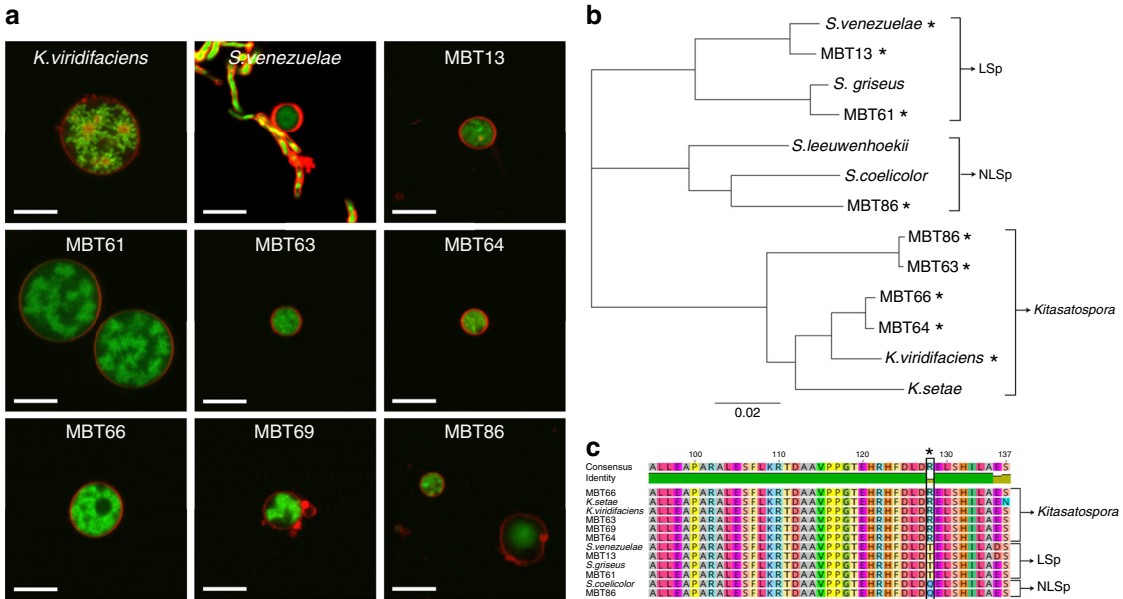

**Fig. 3** Formation of S-cells is widespread in filamentous actinomycetes. **a** Morphology of S-cells released by *K. viridifaciens*, *S. venezuelae* and a number of filamentous actinomycetes from our culture collection (all referred to with the prefix MBT). Cells were stained with FM5-95 (red) and SYTO-9 (green) to visualize membranes and DNA, respectively. **b** Phylogenetic tree of filamentous actinomycetes based on the taxonomic marker *ssgB*. Strains with the ability to form S-cells are indicated with an asterisk (*). *Streptomyces* strains that are able to produce spores in liquid-grown cultures are referred to as LSp (for Liquid Sporulation), while those unable to sporulate in liquid environments are called NLSp (No Liquid Sporulation[26]). This classification is based on amino acid residue 128 in the conserved SsgB protein, which is a threonine (T) or glutamine (Q) for LSp and NLSp strains, respectively. Please note that an arginine (R) is present at this position in all *Kitasatospora* strains (**c**). Scale bars represent 5 μm

*coelicolor*, *Streptomyces lividans*, *Streptomyces griseus* and *Streptomyces venezuelae* as the reference strains. We used a cut-off diameter of 2 μm to distinguish small S-cells from spores. Spherical cells, similar to S-cells were evident in hyperosmotic media in *S. venezuelae* and in 7 out of the 96 wild isolates (Supplementary Fig. 4a). The cells were variable in size within the same strains and between strains (Fig. 3a, Supplementary Table 3) and showed differences in the organization of their DNA (Fig. 3a). No S-cells were found in *S. coelicolor*, *S. griseus* or *S. lividans* under the tested conditions. Phylogenetic analysis based on 16S rRNA (Supplementary Fig. 4b), or the taxonomic marker gene *ssgB* used for classifying morphologically complex actinomycetes[26], revealed that the formation of S-cells is common in at least two genera (Fig. 3b). Moreover, the ability to form S-cells was not restricted to strains that sporulate in liquid-grown cultures. This is based on the observation that MBT86, which is classified as a non-liquid sporulating strain, also generates S-cells (Fig. 3c). Altogether, these results show the ability to generate S-cells, without artificial means such as lysozyme and/or cell wall-targeting antibiotics, is widespread in filamentous actinomycetes.

**S-cells are able to switch to the mycelial mode-of-growth**. To determine where S-cells are generated in the hyphae, we performed live imaging of growing germlings of *K. viridifaciens* (Supplementary Movie 1). Approximately 7 h after the visible emergence of germ tubes, we detected a transient arrest in tip extension of the leading hypha (Fig. 4a, t = 400 min). Shortly thereafter, small S-cells became visible, which were extruded from the hyphal tip (see arrows in Fig. 4a). These cells rapidly increased in size and number. After 545 min, a narrow branch (Fig. 4a, arrowhead) was formed in the apical region from which the S-cells were initially extruded. Subapically, other branches became visible ~210 min after the first appearance of these cells (Supplementary Movie 1, t = 770 min). Notably, such branches frequently also extruded S-cells, similarly to the leading hypha

(Supplementary Movie 2). This showed that S-cells are produced at hyphal tips after apical growth was arrested.

Further characterization of S-cells from *K. viridifaciens* revealed that these cells had a granular appearance and contained membrane assemblies that stained with FM5-95 (Fig. 4b, arrows, Supplementary Movie 3). Notably, these assemblies often co-localized with DNA (Fig. 4b, arrows). To study S-cells in more detail, we separated them from the mycelia after 7 days by filtration (see Methods). In agreement with our previous findings, we also detected agglomerates of membrane assemblies in close proximity of the DNA using electron microscopy analysis (Fig. 4c). Additionally, we noticed that S-cells possessed a disorganized surface, characterized by membrane protrusions that appeared to detach from the S-cells (Fig. 4d, e), and an apparent deficiency in normal cell-wall biogenesis (compare with the cell surface of the hypha in Fig. 4f, g).

To establish if S-cells are truly viable cells, they were plated onto plates supplemented with sucrose. After 7 days of growth, many mycelial colonies were found (±1.6 × 10$^4$ CFUs ml$^{-1}$ of the filtered culture) demonstrating that the cells indeed were viable, and that such cells are only transiently CWD. To exclude that colonies were formed by spores present in the filtrate, we deleted the *ssgB* gene that is required for sporulation[27]. Indeed, this led to a non-sporulating variant of *K. viridifaciens* (Fig. 4h). Like the wild-type strain, the *ssgB* mutant formed CWD S-cells in hyperosmotic growth conditions (Fig. 4i). Time-lapse microscopy (Supplementary Movie 4) revealed that S-cells of the *ssgB* mutant were able to initiate filamentous growth and establish mycelial colonies (Fig. 4j). A switch to mycelial growth was also observed when S-cells were inoculated in liquid medium, whether or not the media was supplemented with high levels of sucrose (data not shown). We noticed that the viability of S-cells was reduced by 60% (decreasing from 1.6 × 10$^4$ to 6.7 × 10$^3$ CFUs ml$^{-1}$) when these cells were diluted in water before plating, consistent with their cell wall deficiency. Microscopy analysis indicated that the surviving S-cells were those that showed abundant staining with

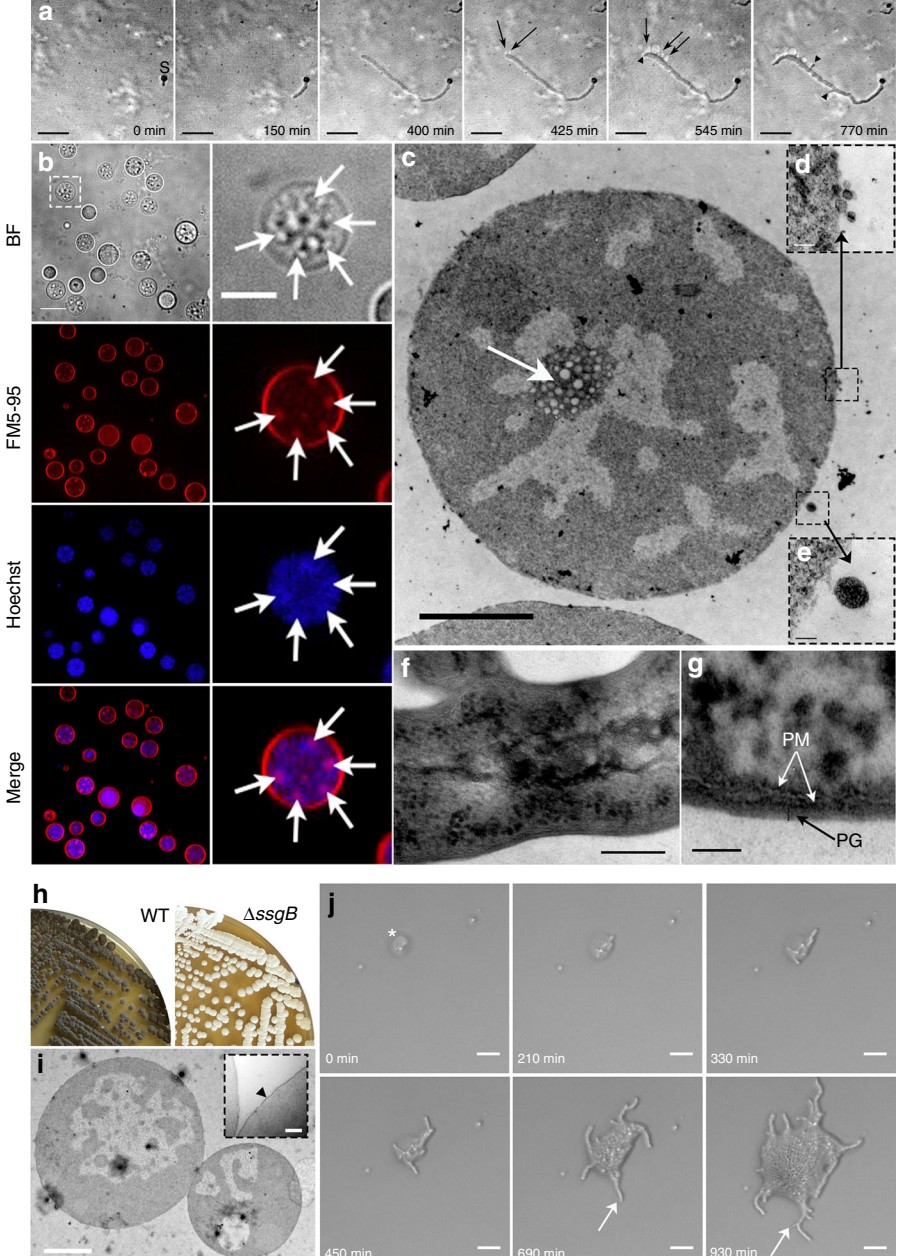

**Fig. 4** S-cells are able to switch to the mycelial mode-of-growth. **a** Time-lapse microscopy stills showing the extrusion of S-cells (arrows) from hyphal tips. The arrowheads indicate new branches, while "S" designates the germinated spore. Images were taken from Supplementary Movie 1 (**b**) Z-stack projection of filtered S-cells (taken from Supplementary Movie 3). Cells were stained with Hoechst and FM5-95 to visualize DNA and membranes, respectively. The intracellular membrane assemblies are indicated with white arrows. **c** Transmission electron micrographs of S-cell reveal the presence of agglomerates of membrane structures (white arrow) in close proximity to the DNA. Contrary to filamentous cells (**f**, **g**), S-cells possess a disorganized cell surface (**d**, **e**). **i** Deletion of *ssgB* in *K. viridifaciens* blocks the formation of grey-pigmented spores (**h**). **i** Transmission electron micrographs indicate that S-cells of the Δ*ssgB* mutant, like those of the wild-type strain, are not surrounded by a cell wall. The arrowhead in the inlay indicates the cell membrane. **j** Time-lapse microscopy stills showing the switch of a Δ*ssgB* mutant S-cell (asterisk) to filamentous growth (arrows). Please note that the mycelial outgrowth leads to the collapse of the S-cell at t = 210 min. Images were taken from Supplementary Movie 4. Scale bars represents 10 μm (**a**, **b**), 5 μm (magnified section in **b**), 2 μm (**c**, **i**), 100 nm (**d**, **e**), 200 nm (**f**, magnified section in **i**), 50 nm (**g**), 20 μm (**j**)

WGA-Oregon (Supplementary Fig. 5). Altogether, these results demonstrate that *K. viridifaciens* generates S-cells that synthesize PG and are able to switch to the mycelial mode-of-growth.

**S-cell formation and switching leads to loss of the KVP1 megaplasmid**. When S-cells were allowed to switch to mycelium on MYM medium, we identified many colonies with developmental defects (Fig. 5). Most obvious was the frequent occurrence of small, brown-pigmented colonies that neither produced aerial hyphae (which are white) nor grey-pigmented spores (Fig. 5c). These brown-colony variants were also observed when protoplasts were plated (Fig. 5b), but were rare when spores were used (Fig. 5a). Such non-differentiating colonies are referred to as bald, for the lack of the fluffy aerial hyphae[28]. To test if this aberrant phenotype was maintained in subsequent generations,

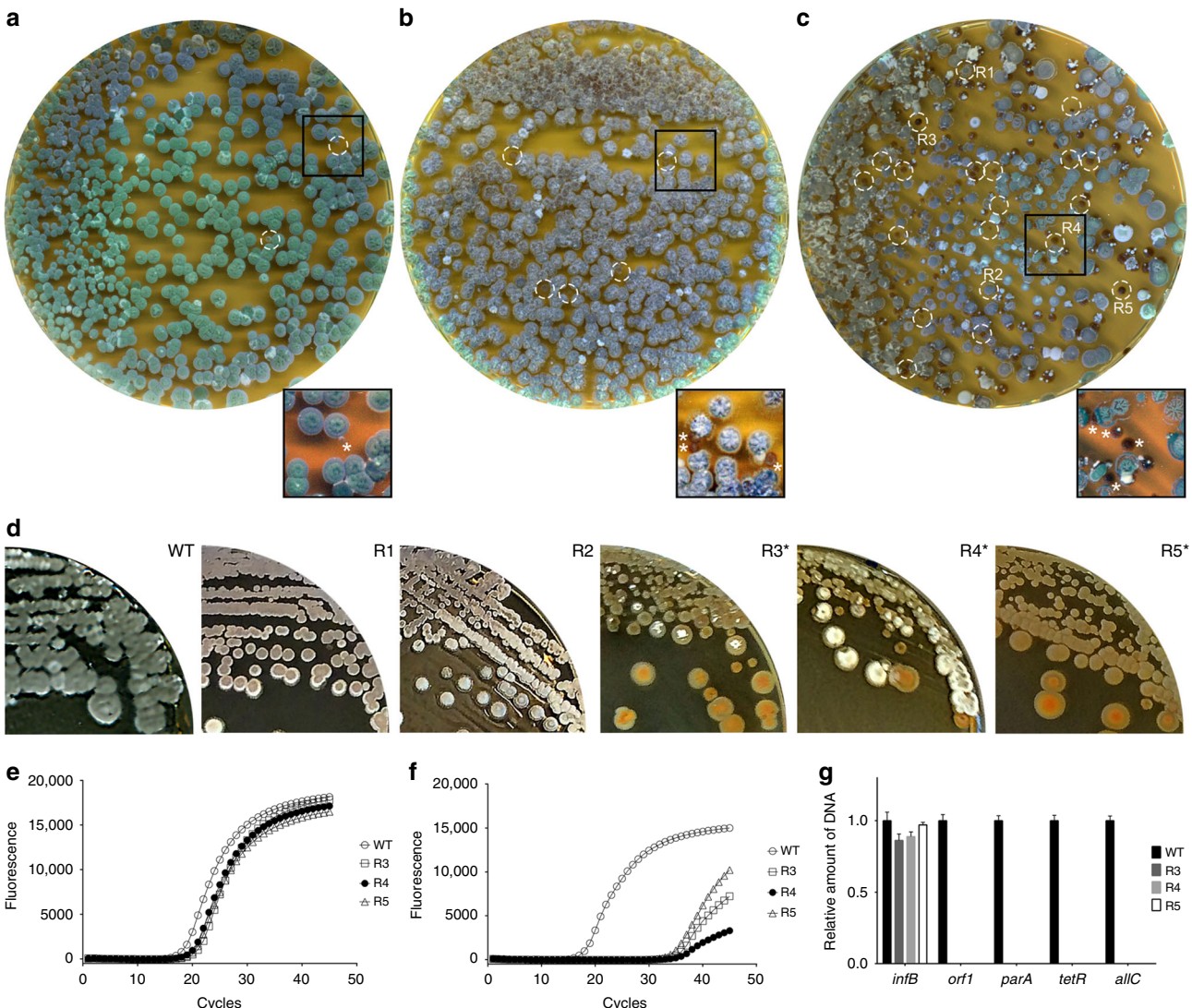

**Fig. 5** S-cell formation and switching leads to loss of the linear megaplasmid KVP1. Morphology of 7-day-old colonies of *K. viridifaciens* on MYM medium obtained after plating spores (**a**), protoplasts (**b**) or S-cells (**c**). The switch of S-cells to the mycelial mode-of-growth yields colonies with different morphologies: besides grey-pigmented colonies (R1, R2), colonies are formed that fail to develop efficiently, and which appear whitish or brown (R3-R5). Brown colonies are also evident when protoplasts are plated (**b**, white circles), but rare when spores are used. **d** Subculturing of R1 and R2 leads to the formation of grey colonies that appear similar to the wild type, while subculturing of R3, R4 and R5 yield colonies that are unable to form a robust sporulating aerial mycelium (brown and white colonies). Quantitative real-time PCR of the *infB* (**e**) and *allC* (**f**) genes using gDNA of the wild type and R3-R5 as the template. In all strains, the *infB* gene located on the chromosome is amplified before the 20th cycle. However, the *allC* gene, located on the KVP1 megaplasmid, is amplified in the wild type before the 20th cycle, but in strains R3-R5 after the 30th cycle. Values represent the average of two replicates. **g** Quantitative comparison of the relative abundance of four megaplasmid genes (*orf1, parA, tetR* and *allC*) and the *infB* gene (located on the chromosome) between the wild type and strains R3-R5. The strong reduction in the abundance of the megaplasmid genes are consistent with loss of this plasmid. qPCR data were normalized to the housekeeping gene *atpD*. Error bars indicate the SEM

we selected three of these bald colonies (R3-R5), and two grey-pigmented colonies with a near wild-type morphology (R1 and R2) for further analysis. The progeny of the grey colonies developed similarly to the wild-type strain, and sporulated abundantly after 7 days of growth (Fig. 5d). In contrast, strains R3-R5 failed to sporulate after 7 days of growth. This phenotype is reminiscent of the defective sporulation seen in colonies of *Streptomyces clavuligerus* that have lost the large linear plasmid pSCL4 following protoplast formation and regeneration[29]. Given that *K. viridifaciens* also contains a large megaplasmid (KVP1[30]), we reasoned that S-cell formation could increase the frequency of the loss of this plasmid. To test this assumption, we performed quantitative real-time PCR using four genes located on the

megaplasmid (*orf1, parA, tetR* and *allC*). Of these, *parA* is implied in plasmid segregation, while *orf1* encodes a plasmid-type DNA replication protein[31]. As a control, we included the housekeeping genes *infB* and *atpD*, which encode the translation initiation factor IF-2 and a subunit of the $F_0F_1$ ATP synthase, respectively. Both of these genes are located on the chromosome. Detectable amplification of *infB* and *atpD* was seen after 19 PCR cycles in strains R3-R5, which was similar to the wild-type strain (Fig. 5e). The same was true for the KVP1-specific genes *orf1, parA, tetR* and *allC* in the wild-type strain. However, amplification of these plasmid marker genes was only seen after 30 PCR cycles in strain R3-R5 (Fig. 5f). This demonstrates that the KVP1-specific genes represented only a very small fraction of the DNA content of

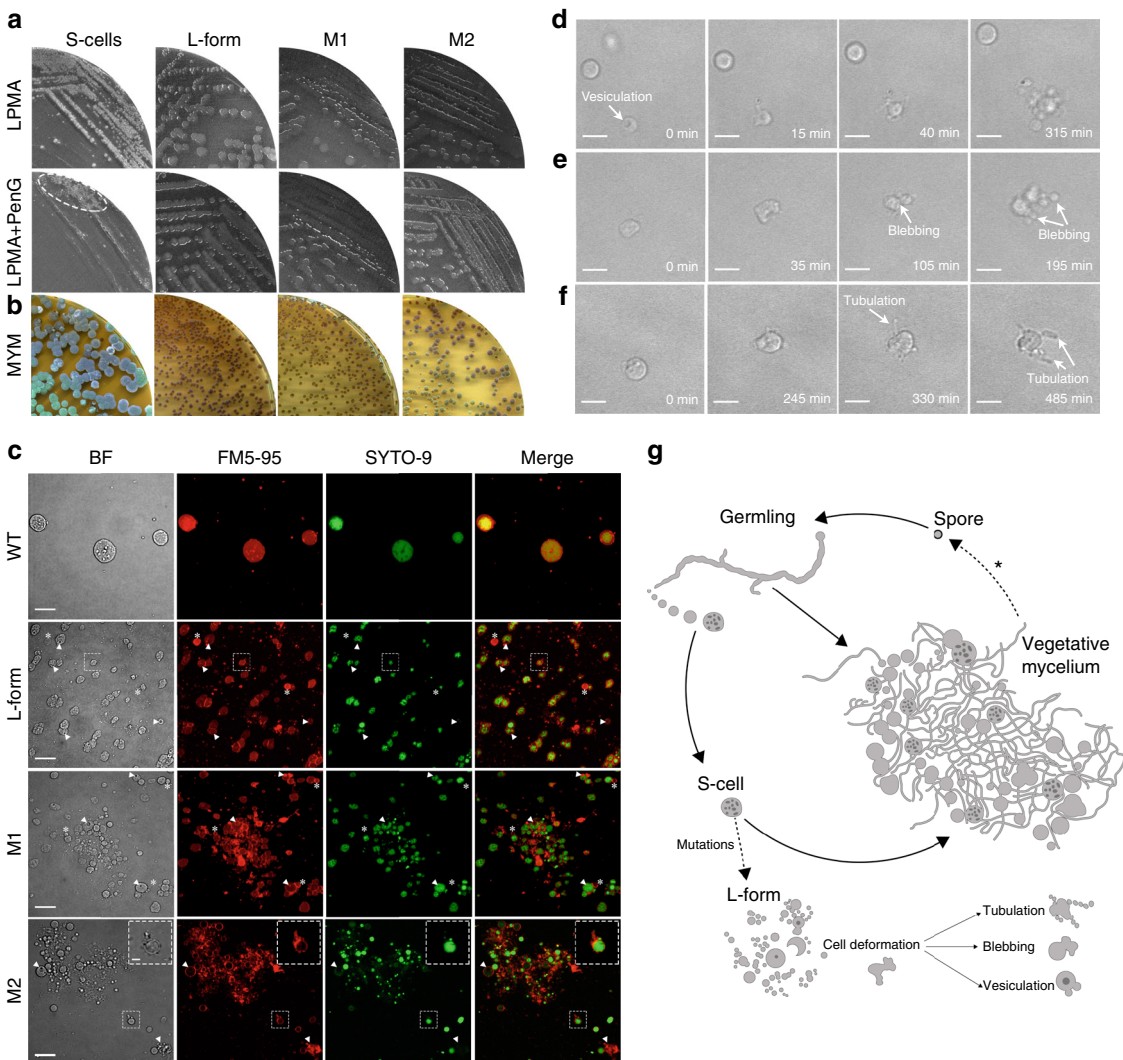

**Fig. 6** Hyperosmotic stress conditions are sufficient to isolate L-form strains. **a** Phenotypic differences are evident when S-cells, penicillin-induced L-forms and cells of the M1 and M2 strains are grown on LPMA medium without (top) and with penicillin (PenG, bottom). Plating S-cells led to the formation of compact mycelial colonies on LPMA medium contrary to the other strains that formed viscous colonies. The addition of 0.6 mg ml$^{-1}$ penicillin abolished efficient switching of S-cells to the mycelial mode-of-growth. Small mycelial colonies derived from S-cells were only formed in the least diluted part (indicated with a dashed ellipse). In contrast, the addition of penicillin neither effected growth of the penicillin-induced L-forms, nor that of strains M1 and M2. **b** All cell wall-deficient cells were able to form mycelial colonies on MYM medium lacking high levels of osmolytes. Unlike the majority of colonies derived from S-cells, the penicillin-induced L-forms and strains M1 and M2 exclusively formed (brown) non-sporulating colonies. **c** Morphology of S-cells in comparison to cells of the penicillin-induced L-form strain and strains M1 and M2 grown for 48 h in LPB medium. Cells were stained with FM5-95 and SYTO-9 to visualize membranes and DNA, respectively. The arrowheads indicate intracellular vesicles, while empty vesicles are indicated with an asterisk. The inlay in M2 shows a proliferation-associated tubulation event. **d-f** Frames from time-lapse microscopy show L-form-like proliferation involving (**d**) vesiculation, (**e**) blebbing and (**f**) membrane tubulation. **g** Formation of S-cells upon prolonged exposure to hyperosmotic stress. Germination of spores under hyperosmotic stress conditions generates germlings, which are able to extrude S-cells. These S-cells are able to switch to the mycelial mode-of-growth, or sporadically acquire mutations that allow them to proliferate like L-forms, which is characterized by tubulation, blebbing or vesiculation. Scale bar represents 10 μm (**c**), 2 μm (inlay panel **c**) or 5 μm (**d-f**)

R3-R5 strains (at least $10^4$ times less abundant than the chromosomal genes *infB* and *atpD*) (Fig. 5g). This is consistent with loss of KVP1 in the mycelial colonies obtained after S-cell switching.

**Prolonged hyperosmotic stress converts S-cells into L-forms.** Although the switch to mycelial growth was exclusively observed when young S-cells were cultured in fresh media, we noticed a dramatic change when S-cells had been exposed for prolonged periods to the hyperosmotic stress conditions. In 9 out of 15 independent experiments, we found that S-cells switched to

mycelial growth, while four times S-cells failed to form a growing culture. Striking, however, were the two independent occasions where S-cells had proliferated in an apparent CWD state. On solid LPMA media, these two independent cell lines, called M1 and M2 (for mutants 1 and 2, respectively, see below), formed viscous colonies, which were morphologically similar to those from an L-form lineage induced by the addition of penicillin. Conversely, filtered S-cells formed compact mycelial colonies on LPMA medium (Fig. 6a). This interesting difference between S-cells and the M1 and M2 lineages suggested that the S-cells represent a transient form of CWD cells, as they formed mycelial colonies even when osmoprotection was available. Consistent

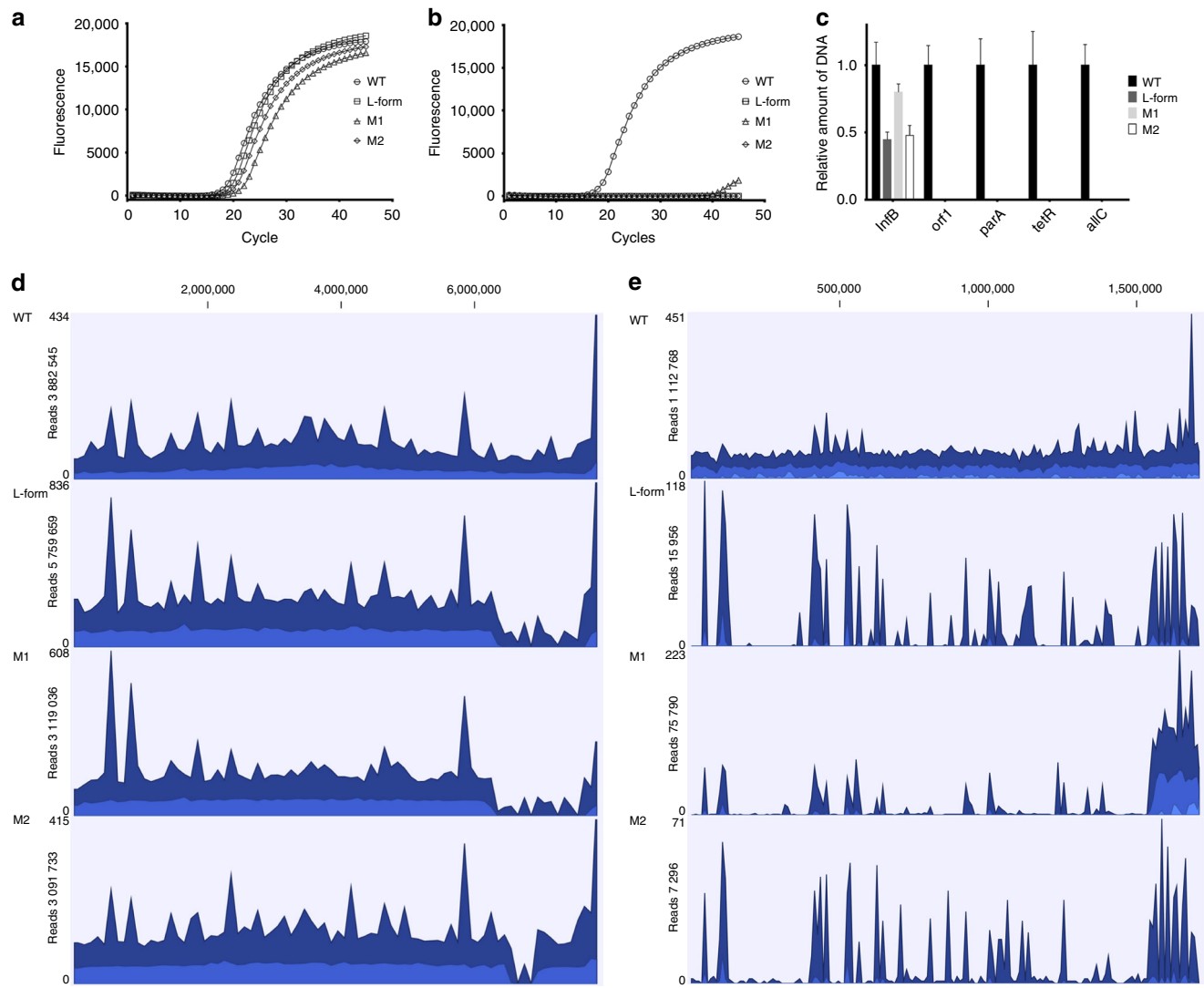

**Fig. 7** Strains proliferating as L-forms lack the KVP1 megaplasmid. Quantitative real-time PCR of the *infB* (**a**) and *allC* (**b**) genes using gDNA of the wild-type strain, the penicillin-induced L-form strain, and strains M1 and M2 that proliferate as L-forms. In all strains, the *infB* gene located on the chromosome is amplified before the 20th cycle. However, the *allC* gene, located on the KVP1 megaplasmid, is only amplified in the wild type strain, but not in any of the other strains. Values represent the average of two replicates. **c** Quantitative comparison of the relative abundance of four megaplasmid genes (*orf1*, *parA*, *tetR* and *allC*) and the *infB* gene (located on the chromosome) between wild-type and L-form strains. qPCR data was normalized to the housekeeping gene *atpD*. Error bars indicate the SEMn. **d–e** Alignment of Illumina reads from the *K. viridifaciens* wild type strain, the penicillin-induced L-form strain, and the M1 and M2 strains against the chromosome (**d**) and the KVP1 megaplasmid (**e**). For all strains, an overall high coverage was observed when reads were aligned against the chromosome (**d**). In contrast, a low coverage was observed when reads were aligned against the megaplasmid KVP1, the wild-type strain being the exception (**e**). Please note that the number of reads corresponding to some parts in the right arms of the chromosomes of the penicillin-induced L-form strain and the M1 and M2 strains is strongly reduced, suggesting major lesions

with this observation, we observed that the addition of 0.6 mg ml$^{-1}$ penicillin inhibited the switch of S-cells to mycelial colonies (Fig. 6a). However, the M1 and M2 strains and the L-form lineage were unaffected by the addition of penicillin, and formed viscous colonies composed of spherical cells in the presence of the antibiotic. Remarkably, switched S-cells, but also strains M1, M2 as well as the penicillin-induced L-form lineage were viable on MYM agar plates lacking osmoprotectant (Fig. 6b). Under these conditions, all strains had formed mycelial colonies and exclusively grew in a filamentous manner.

Also, liquid-grown cultures of M1 and M2 exclusively consisted of CWD cells when sucrose and MgCl$_2$ were added (Fig. 6c, BF panels). The spherical cells produced by M1 and M2 were comparable in size to the penicillin-induced L-forms. Further microscopic analysis revealed that the cells from M1

and M2 contained inner vesicles (arrowheads in Fig. 6c) and tubular protrusions emerging from the cell surface (Fig. 6c, inlay). The vast majority of cells contained DNA, although some empty vesicles were also evident in M1 and M2 (Fig. 6c, asterisks). Time-lapse microscopy revealed that both strains proliferated, whereby smaller progeny cells were released following deformation of the mother cell membrane by either vesiculation (Fig. 6d, taken from Supplementary Movie 5), blebbing (Fig. 6e, taken from Supplementary Movie 6) or tubulation (Fig. 6f, taken from Supplementary Movie 7). Altogether these results inferred that strains M1 and M2 morphologically closely resemble L-forms, both in their ability to proliferate in the CWD state, and the capacity to grow in the presence of penicillin (see above). However, instead of originating from prolonged exposure to antibiotic and/or

lysozyme treatment, they originate from hyperosmotically stress-induced cells.

The low frequency at which the M1 and M2 L-form cell lines had been obtained suggests that M1 and M2 had acquired mutations that enabled these strains to proliferate without a proper cell wall. Real-time qPCR studies revealed that M1 and M2, as well as the penicillin-induced L-form cell line, appeared to have lost the megaplasmid genes *tetR*, *allC*, *orf1* and *parA* (Fig. 7). In agreement, Illumina sequencing revealed a low coverage of KVP1-located sequences (Fig. 7e). However, loss of the mega-plasmid is not sufficient to drive the transition from S-cells to L-forms, as strains R3-R5, all of which had also lost the KVP1 megaplasmid, formed mycelia extruding S-cells under hyper-osmotic stress conditions (data not shown). Further analyses indicated that M1 and M2 had acquired several other mutations, including both major lesions in the right arm of the chromosome (Fig. 7d, e) and a number of point mutations (Supplementary Table 4, 5). Interestingly, both strains carried a mutation in the gene BOQ63_RS21920 that encodes a putative metal ABC transporter. Transporters are often used to cope with osmotic stress conditions[32]. We also identified mutations in the penicillin-induced L-form strain (Supplementary Table 6). These mutations, however, differed from those observed in the hyperosmotically-induced L-form strains M1 and M2. Notably, the mutations in the penicillin-induced L-form appeared to directly relate to cell wall biogenesis, for example, in the case of the mutation in *uppP*. The encoded protein is involved in the recycling pathway of the carrier lipid undecaprenyl phosphate (BOQ63_RS22750) that transports glycan biosynthetic intermediates for cell wall synthesis. Although it is currently not known how all these mutations affect morphology, our results demonstrate that prolonged exposures to hyperosmotic stress are mutagenic conditions through which a filamentous bacterium can be converted into an L-form mutant strain that proliferates without the cell wall.

## Discussion

Filamentous actinomycetes have been intensely studied for more than 50 years as a model for bacterial development. Here, we provide compelling evidence that S-cells represent a natural and previously unnoticed developmental stage in these organisms, when they are exposed to hyperosmotic stress conditions (Fig. 6g). These S-cells are extruded from the hyphal tips, containing DNA, and are viable with the ability to grow into mycelial colonies. Furthermore, upon prolonged exposure to hyperosmotic stress, S-cells may also accumulate mutations that enable them to efficiently proliferate in the wall-deficient state we have dubbed hyperosmotically-induced-L-forms. Our data show that these L-forms can simply emerge as the product of prolonged exposure of cells to hyperosmotic conditions, without directly requiring cell wall-targeting agents. This work thus provides leads towards dissecting the ecological relevance that such cells may have.

Environmental fluctuations can dramatically influence the availability of water in ecosystems and present osmotic shock conditions to organisms. For instance, microorganisms living in hyperarid regions or hypersaline aquatic environments are frequently exposed to desiccation or hypertonicity[33]. Also, microbes in snow and ice habitats experience low water availability and hypersaline or hyper-acidic environments[34]. Bacteria can adapt to these fluctuations by modulating fatty acid synthesis, accumulating or synthesizing osmoprotectants, protecting their DNA, and secreting extracellular polymeric substance[33,35].

Here, we focused on the adaptation of filamentous actinomycetes, which are common in any soil, to extended periods of hyperosmotic stress. As expected, we detected that these bacteria increased the amount of membrane in the hyphae and condensed

their nucleoids. A surprising discovery was the extrusion of CWD S-cells. Together with sporulation and the recently discovered explorative mode-of-growth[36], the ability to form S-cells extends the repertoire by which filamentous actinomycetes can thrive in changing environments. Our work reveals that the ability to extrude S-cells is common in filamentous actinomycetes and occurs in both *Streptomyces* and *Kitasatospora* species. We used a stringent selection to assess S-cell formation, which we define as CWD cells that contain DNA and that are larger in size than 2 μm (to exclude that these are swollen spores). In this study, strains were solely screened in liquid-grown cultures in hyperosmotic stress conditions. Being this stringent, we anticipate that potentially many more strains are able to make S-cells under influence of other stresses. For instance, it is known that cell wall deficiency is stimulated by hypoxic, temperature or nutrient stresses[37,38]. These are also conditions that filamentous actinomycetes are frequently exposed to in heterogenous soil environments.

S-cells are only transiently CWD and have the ability to switch to the mycelial mode-of-growth. Strikingly, many of the switched colonies appear to have developmental defects (see Fig. 5c). These developmental phenotypes are reminiscent of those that have been associated with genetic instability in a range of actinomycetes[39,40]. Here, we demonstrated that colonies that were unable to establish reproductive aerial hyphae lacked the KVP1 megaplasmid. This is likely caused by initial differences in the amount and DNA content between S-cells. While the majority of cells will receive one or more KVP1 copies during their formation, a fraction of cells will receive none. Additionally, S-cells may also carry different numbers of chromosomes, based on the range of sizes that S-cells have. Such multinucleated cells are prone to recombination events[41,42], which will furthermore increase diversity. Altogether, we think that these differences in DNA content in S-cells contributes to the morphological heterogeneity observed in the mycelial colonies derived from S-cells.

In addition to switching to mycelial growth, S-cells can have several other fates. As these cells are wall-deficient, they are prone to lysis due to influx of water. Indeed, exposure to water leads to a steep decline in their ability to outgrow into colonies. However, when S-cells lyse, the DNA cargo will be released into the environment. Given the large number of biosynthetic gene clusters (BGCs) that are present in the genomes of filamentous actinomycetes, including their resistance determinants, this release of DNA may be a significant, and previously unknown mechanism by which resistance genes are spread. In contrast to releasing DNA into the environment, we speculate that S-cells may also take up DNA, similar to other CWD cell types, such as protoplasts or L-forms[43]. Whether S-cells play a role in horizontal gene transfer is under current investigation.

Our work shows that S-cells are extruded from hyphal tips into the environment, coinciding with an arrest in tip growth. Following their release, the extruding hypha reinitiates growth, indicating that the extrusion process occurs in a manner that apparently is not lethal for the filament from which the cells are released. Tip growth in filamentous actinomycetes is coordinated by the polarisome complex, of which the DivIVA protein is a crucial member[44]. Recent work revealed that hyperosmotic stress has a dramatic effect on the polar growth machinery. Following osmotic upshift experiments, tip growth is arrested, followed by relocation of the apical growth machinery to subapical sites. As a consequence, lateral branches emerge from the leading hyphae[24]. We hypothesize that an imbalance between cell wall synthesis and cell wall turnover could locally lead to changes in the thickness or structure of the cell wall, allowing S-cells to escape from the sacculus.

L-forms have been studied for many decades, and only recently are we beginning to understand their exciting biology, especially due to groundbreaking work from the Errington lab. L-form cells have been artificially generated from many different bacteria in many laboratories, invariably aimed at targeting the biosynthesis pathway of the cell wall. To that end, cells are typically exposed to high levels of antibiotics, either or not combined with lysozyme treatment[18,23]. Our work expands on this research by providing for the first-time evidence that CWD strains can emerge solely by exposure to hyperosmotic stress conditions and implies an environmental relevance of this cell type. A crucial and limiting step in the formation of L-forms in *B. subtilis*, as well as in other bacteria, is the escape of a protoplast from the cell-wall sacculus. This process requires lytic activity, which usually comes from lysozyme activity[45]. Our data show that actinomycetes have a natural ability to release such CWD cells when exposed to hyperosmotic conditions. Under prolonged exposure to osmotic stress, some cells are able to acquire mutations allowing these cells to propagate as L-forms. In line with these findings, recent work shows that *B. subtilis* and *S. aureus* both are able to convert to wall-deficient cells[45]. This has been shown in an animal infection model as well as in macrophages, where lysozyme activity from the host converts walled bacteria into CWD cells. Collectively, these results indicate that CWD cells represent an adaptive morphology allowing cells to overcome environmental challenges, such as antibiotic treatment or hyperosmotic stress conditions.

In summary, our work provides evidence for a new, CWD cell type in the biology of filamentous actinomycetes. It further expands the large diversity in bacterial cell types, and the plasticity that microorganisms employ to handle environmental stresses. It remains to be elucidated how the ability to form S-cells improves fitness in these filamentous actinomycetes, and how this morphogenetic switch is regulated.

## Methods

**Strains and media**. Bacterial strains used in this study are shown in Supplementary Table 7. To obtain sporulating cultures, *Streptomyces* and *Kitasatospora* species were grown at 30 °C for 4 days on MYM medium[46]. To support growth of CWD cells, strains were grown on solid medium L-Phase Medium (LPMA), containing 0.5% glucose, 0.5% yeast extract, 0.5% peptone, 20% sucrose, 0.01% MgSO$_4$·7H$_2$O and 0.75% Iberian agar (all w/v). After autoclaving, the medium was supplemented with MgCl$_2$ (final concentration of 25 mM) and 5% (v/v) horse serum.

L-phase broth (LPB) was used as liquid medium to support growth of wall-deficient cells. LPB contains 0.15% yeast extract, 0.25% bacto-peptone, 0.15% oxoid malt extract, 0.5% glucose, 0.64 M sucrose, 1.5% oxoid tryptic soy broth powder (all w/v) and 25 mM MgCl$_2$. To test the effect of different sucrose concentrations on mycelial growth and the formation of S-cells, the amount of sucrose in LPB was changed to obtain final concentrations of 0.0, 0.06, 0.18, 0.50 and 0.64 M. The influence of other osmolytes was analysed by replacing sucrose with NaCl (0.6 M) or sorbitol (1 M). In total, 50 ml cultures were inoculated with 10$^6$ spores ml$^{-1}$ and grown in 100 ml flasks. Cultures were incubated at 30 °C, while shaking at 100 rpm.

To prepare protoplasts of *K. viridifaciens*, the wild-type strain was grown for 48 h in a mixture of TSBS and YEME (1:1 v/v) supplemented with 5 mM MgCl$_2$ and 0.5% glycine. Protoplasts were prepared by incubating the mycelium for three hours in 10 mg ml$^{-1}$ lysozyme solution[47]. Freshly made protoplasts were diluted and immediately used for fluorescence microscopy.

**Optical density measurements**. The growth of *K. viridifaciens* was monitored with the Bioscreen C reader system (Oy Growth Curves AB Ltd). To this end, aliquots of 100 µl of LPB medium with different concentrations of sucrose were added to each well of the honeycomb microplate and inoculated with 10$^7$ spores ml$^{-1}$. Growth was monitored for 24 h at 30 °C, while shaking continuously at medium speed. The OD wide band was measured every 30 min and corrected for the absorbance of liquid medium without inoculum. In total, five replicate cultures were used for each osmolyte concentration. The effect of sodium chloride and sorbitol as osmolyte were tested using the same procedure, with the differences that the final volume of the cultures was 300 µl, and the experiment was run for 96 h.

**Quantification of the number and size of colonies**. Serial dilutions of *K. viridifaciens* spores were plated in triplicate in LPMA (high osmolarity) and LPMA without sucrose, MgCl$_2$ and horse serum (low osmolarity). After 7 days of

incubation at 30 °C, the number of colonies was counted to determine the CFU ml$^{-1}$. Quantification of the surface area of colonies was done with FIJI[48].

**Screening for strains with the ability to release S-cells**. To identify strains that are able to release S-cells, strains from an in-house culture collection[25] were initially grown in flat-bottom polysterene 96-well plates, of which each well contained 200 µl LPB medium and 5 µl of spores. The 96-well plate was sealed with parafilm and incubated at 30 °C for 7 days. The cultures were then analysed with light microscopy, and strains with the ability to release S-cells with a diameter larger 2 µm were selected. The selected strains were then grown in 250 mL flasks containing 50 mL LPB medium (10$^6$ spores ml$^{-1}$) at 30 °C while shaking at 100 rpm. After 7 days, aliquots of 50 µl of the bacterial cultures were fluorescently stained with SYTO-9 and FM5-95. The surface area of the S-cells was determined in FIJI[48]. Assuming circularity of these cells, the corresponding diameter D was then calculated as $D = 2 * \sqrt{\left(\frac{area}{\pi}\right)}$

**Filtration of S-cells from *K. viridifaciens***. In total, 50 ml of LPB cultures of *K. viridifaciens*, inoculated with 10$^6$ spores ml$^{-1}$, were grown for 2 or 7 days at 30 °C in an orbital shaker at 100 rpm. To separate the S-cells from the mycelium, the cultures were passed through a sterile filter made from an EcoCloth™ wiper. A subsequent filtration step was done by passing the S-cells through a 5 µm Isopore™ membrane filter. The filtered vesicles were centrifuged at 190 g for 40 min, after which the supernatant was carefully removed with a 10 mL pipette to avoid disturbance of the S-cells. Same procedure was followed to filtrate S-cells from ΔssgB mutant, although the cultures were inoculated with an individual colony that had been grown on MYM medium for 6 days.

**Viability and subculturing of S-cells from *K. viridifaciens***. To verify the viability of S-cells, the filtered cells were directly plated or incubated in 10 mg ml$^{-1}$ lysozyme solution[47] for 3 h at 30 °C, while shaking at 100 rpm. The filtered S-cells were then centrifuged at 190 g for 40 min and resuspended in one volume of fresh LPB. Serial dilutions of the S-cells in LPB or water were then plated, in triplicate, on LPMA or MYM medium. The plates were grown for 7 days at 30 °C, and the CFU values were determined for each treatment.

**Generation of the penicillin-induced L-form cell line**. Generation of the *K. viridifaciens* L-form lineage was performed by inoculating the wild-type strain in 50 mL LPB medium, supplemented with lysozyme and/or penicillin G (Sigma), in 100 mL flasks in an orbital shaker at 100 rpm. Every week, 1 mL of this culture was transferred to fresh LPB medium[19]. After the 8th subculture, the inducers were removed from the cultivation medium and the obtained lineage did not revert back to the walled state on LPMA plates or in LPB medium. A single colony obtained after the 8th subculture was designated as penicillin-induced L-forms.

**Construction of the *ssgB* deletion construct pKR1**. The *ssgB* (BOQ63_RS34980) mutant was created in *K. viridifaciens* using pKR1, which is a derivative of the unstable plasmid pWHM3 as described[49]. In the *ssgB* mutant, nucleotides +20 to +261 relative to the start codon of *ssgB* were replaced with the *loxP-apra*-resistant cassette[50].

**Phylogenetic analysis**. The 16S rRNA sequences from strains of the in-house culture collection were previously determined[25]. Homologues of *ssgB* in these strains were identified by BLAST analysis using the *ssgB* sequence from *S. coelicolor* (SCO1541) as the input. For the *Streptomyces* and *Kitasatospora* strains whose genome sequence was not available, the *ssgB* sequence was obtained by PCR with the *ssgB* consensus primers (Supplementary Table 8). Geneious 9.1.7 was used to make alignments of *ssgB* and 16S rRNA, and for constructing neighbour-joining trees.

**Quantitative real-time PCR**. Filtered S-cells were allowed to regenerate on MYM medium, from which three regenerated bald colonies (R3, R4 and R5) were selected. After two rounds of growth on MYM, bald colonies of the three strains were grown in TSBS for 2 days at 30 °C, and genomic DNA was isolated from these strains[47]. Primers were designed to amplify the *infB* (BOQ63_RS29885) and *atpD* (BOQ63_RS19395) genes located in the chromosome, and four genes located on the KVP1 megaplasmid: *allC* (BOQ63_RS01235), *tetR* (BOQ63_RS02930), *parA* (BOQ63_RS04095) and *orf1* (BOQ63_RS04285) (Supplementary Table 8). The PCR reactions were performed in duplicate in accordance with the manufacturer's instructions, using 5 ng of DNA, 5% DMSO and the iTaq Universal SYBR Green Supermix Mix (Bio-Rad). Quantitative real-time PCR was performed using a CFX96 Touch Real-Time PCR Detection System (Bio-Rad). To normalize the relative amount of DNA, the wild-type strain was used as a control, using the *atpD* gene as a reference.

**Isolation of the L-form cell lines M1 and M2**. Fifteen replicate cultures of *K. viridifaciens* were grown for 7 days in LPB medium. After filtration, the S-cells were transferred to fresh LPB medium. The cultures that had not switched to

mycelium after 3 days of cultivation were kept for further analysis. Two cultures turned dark green after 7 days, which after inspection with light microscopy contained proliferating L-form cells. These cell lines were named M1 and M2.

**Microscopy**. Bright-field images were taken with the Zeiss Axio Lab A1 upright Microscope, equipped with an Axiocam MRc with a resolution of 64.5 nm/pixel. Fluorescent dyes (Molecular Probes[TM]) were added directly to 100 μl aliquots of liquid-grown cultures. For visualization of membranes, FM5-95 was used at a final concentration of 0.02 mg ml$^{-1}$. Nucleic acids were stained with 0.5 μM of SYTO-9 or 0.05 mg ml$^{-1}$ of Hoechst 34580. The detection of PG was done using 0.02 mg ml$^{-1}$ Wheat Germ Agglutinin (WGA) Oregon Green or 1 μg ml$^{-1}$ BOPIPY FL vancomycin (which stains nascent PG). Prior to visualization, cells and mycelium were applied on a thin layer of LPMA (without horse serum) covering the glass slides. Confocal microscopy was performed using a Zeiss Axio Imager M1 Microscope. Samples were excited using a 488-nm laser, and fluorescence emissions for SYTO-9 and WGA Oregon Green were monitored in the region between 505–600 nm, while a 560 nm long pass filter was used to detect FM5-95.

The characterization of the membrane assemblies in S-cells was done on a Nikon Eclipse Ti-E inverted microscope equipped with a confocal spinning disk unit (CSU-X1) operated at 10,000 rpm (Yokogawa, Japan) using a 100x Plan Fluor Lens (Nikon, Japan) and illuminated in brightfield and fluorescence. Samples were excited at wavelengths of 405 and 561 nm for Hoechst and FM5-95, respectively. Fluorescence images were created with a 435 nm long pass filter for Hoechst, and 590–650 nm band pass for FM5-95. Z-stacks shown in Supplementary Movie 3 were acquired at 0.2 μm intervals using a NI-DAQ controlled Piezo element.

Visualization of stained CWD cells for size measurements were done using the Zeiss Axio Observer Z1 microscope. Aliquots of 100 μl of stained cells were deposited in each well of the ibiTreat μ-slide chamber (ibidi®). Samples were excited with laser light at wavelengths of 488, the green fluorescence (SYTO-9, BODIPY FL vancomycin, WGA-Oregon) images were created with the 505–550 nm band pass, while a 650 nm long pass filter was used to detect FM-595.

**Time-lapse imaging**. To visualize the emergence of S-cells, spores of *K. viridifaciens* were pre-germinated in TSBS medium for 5 h. An aliquot of 10 μl of the recovered germlings was placed on the bottom of an ibiTreat 35 mm low imaging dish (ibidi®), after which an LPMA patch was placed on top of the germlings.

To visualize switching, the S-cells produced by the ΔssgB mutant were collected after 7 days by filtration from a liquid-grown culture. A 50 μl aliquot of the filtrate was placed on the bottom of an ibiTreat 35 mm low imaging dish (ibidi®) with a patch of R5 on top.

To visualize the proliferation of M1 and M2, the strains were grown for 48 h in LPB. Aliquots of the culture were collected and centrifuged at 7516 g for 1 min, after which the supernatant was removed, and the cells resuspended in fresh LPB. Serial dilutions of the cells were placed in wells of an ibiTreat μ-slide chamber (ibidi®).

All samples were imaged for ~15 h using an inverted Zeiss Axio Observer Z1 microscope equipped with a Temp Module S (PECON) stage-top set to 30 °C. Z-stacks with a 1 μm spacing were taken every 5 min using a 40x water immersion objective. Average intensity projections of the in-focus frames were used to compile the final movies. Light intensity over time was equalised using the correct bleach plugin of FIJI.

**Electron microscopy**. To visualize the vegetative mycelium of *K. viridifaciens* by transmission electron microscopy (TEM), the strain was grown in TSBS medium for 48 h. An aliquot of 1.5 ml of the culture was centrifuged for 10 min at 190 g, after which the supernatant was carefully removed with a pipette. The mycelium was washed with 1X PBS prior to fixation with 1.5% glutaraldehyde for 1 h at room temperature. The fixed mycelium was centrifuged with 2% low melting point agarose. The solid agarose containing the embedded mycelium was sectioned in 1 mm³ blocks, which were post-fixed with 1% osmium tetroxide for 1 h. The samples were then dehydrated by passing through an ethanol gradient (70, 80, 90 and 100%, 15 min per step). After incubation in 100% ethanol, samples were placed in propylene oxide for 15 min followed by incubation in a mixture of Epon and propylene oxide (1:1) and pure Epon (each step 1 h). Finally, the samples were embedded in Epon and sectioned into 70 nm slices, which were placed on 200-mesh copper grids. Samples were stained using uranyl-430 acetate (2%) and lead-citrate (0.4%), if necessary, and imaged at 70 kV in a Jeol 1010 transmission electron microscope.

To image S-cells, cultures of the *K. viridifaciens* wild-type and the ΔssgB mutant strains that had been grown in LPB medium for 7 days were immediately fixed for 1 h with 1.5% glutaraldehyde. Filtered S-cells (see above) were then washed twice with 1X PBS prior to embedding them in 2% low melting agarose. A post-fixation step with 1% OsO₄ was performed before samples were embedded in Epon and sectioned into 70 nm slices (as described above). Samples were stained using uranyl-430 acetate (2%) and lead-citrate (0.4%), if necessary, and imaged at 70 kV in a Jeol 1010 transmission electron microscope

**Image analysis**. Image analysis was performed using the FIJI software package. To describe the morphological changes during hyperosmotic stress, we compared

mycelium grown in LPB with or without 0.64 M of sucrose (i.e., the concentration in LPB medium). After making average Z-stack projections from mycelia, 10 hyphae derived from independent mycelia projections were further analysed. For each hypha, the total length was measured using the segmented line tool (Supplementary Fig. 1a, h) and the number of branches (asterisks in Supplementary Fig. 1a, h) emerging from that hypha was counted. The hyphal branching ratio was calculated as the number of branches per micrometer of leading hypha.

To calculate the surface area occupied by membrane in hyphae either or not exposed to 0.64 M sucrose, we divided the total surface area (Supplementary Fig. 1e, l) that stained with FM5-95 by the total surface area of the hypha (Supplementary Fig. 1c, j). FIJI was also used to measure the average surface area of the nucleoid (using SYTO-9 staining, Supplementary Fig. 1g, n) in both growth conditions. Student's T-tests with two-sample unequal variance were performed to calculate P-values and to discriminate between the samples.

To determine the size of CWD cells, we compared cells of penicillin-induced L-form to fresh protoplasts and S-cells, all obtained or prepared after 48 h of growth. Cells were stained with FM5-95 and SYTO-9 and deposited in the wells of an ibiTreat μ-slide chamber (ibidi®). The size of the spherical was determined as the surface area enclosed by the FM5-95-stained membrane. For the particular case of L-forms, where empty vesicles are frequent, only cells that contained DNA were measured. At least 200 cells of each CWD variant were analysed. Proliferating L-forms in which the mother cell could not be separated from the progeny, were counted as one cell.

**Genome sequencing and SNP analysis**. Whole-genome sequencing followed by de novo assembly (Illumina and PacBio) and variant calling analyses were performed by BaseClear (Leiden, The Netherlands). The unique mutations were identified by direct comparison to the parental strain *Kitasatospora viridifaciens* DSM40239 (GenBank accession number PRJNA353578[30]). The single and multiple nucleotide variations were identified using a minimum sequencing coverage of 50 and a variant frequency of 70%. To reduce the false positives the initial variation list was filtered, and the genes with unique mutations were further analysed. All variants were verified by sequencing PCR fragments (primer sequences in Supplementary Table 8).

**Alignment of Illumina sequences**. Alignments of Illumina reads were performed using CLC Genomics Workbench 8.5.1 (Qiagen, the Netherlands). Raw Illumina (Hiseq2500 system) sequences of the wild-type, penicillin-induced L-form and M1 and M2 strains were imported and mapped to the reference genome of *K. viridifaciens* DSM40239 (GenBank sequence MPLE00000000.1) through the "Map reads to reference" function in the NGS core tools. Mismatch cost was set to two and non-specific matches were handled by mapping them randomly.

## Data availability

Genomic sequence data for the mutant strains have been deposited in the NCBI SRA database under accession codes SAMN10407336 to SAMN10407338. Other data that support the findings of this study are available in this article and its Supplementary Information files, or from the corresponding author upon request.

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

## Acknowledgements

We are indebted to Mark Leaver, Roberto Kolter, Danny Rozen, Ben Lugtenberg and Paul Hooykaas for critical reading of the manuscript. This work was supported by a VIDI grant (12957) from the Dutch Applied Research Council to D.C.

## Author contributions

K.R., E.U., J.W., Z.Z., A.J.W., A.M., D.H., A.B., G.P.W. and D.C. collected the data and aided in data analysis. K.R., G.P.W. and D.C. designed the experiments, while D.C. supervised the research. K.R., G.P.W. and D.C. wrote the paper with input from all co-authors.

## Additional information

**Competing interests:** The authors declare no competing interests.

