## [Peer Review File · Nature Communications]

Reviewers' comments:

Reviewer #1 (Remarks to the Author):

This a fascinating study and although I have no issues with the main findings, there are some aspects of the manuscript that require attention.

Abstract:

It is an over-interpretation to state that this study provides important insights into how genes may disseminate in the environment. This has not been addressed experimentally and is pure speculation. I recommend removing it from the Abstract, but it is worthwhile mentioning in the Discussion.

Results:

Loss of KVP1 - the developmental phenotypes described are very reminiscent of the results of genetic instability in *Streptomyces*, due to loss of large amounts of the terminal regions of the chromosome. Loss of plasmid could be verified by PFGE and, at the same time, this would permit analysis of the chromosome. The results of genome sequencing could also confirm the nature of the genetic changes, although only results of a few SNPs are reported. Does the sequence analysis confirm the complete absence of the megaplasmid? Is there evidence for loss of any other genes?

Reviewer #2 (Remarks to the Author):

The work of Ramijan and colleagues describes the intriguing observation that some sporulating actinobacteria grown under high osmotic conditions extrude membrane vesicles of varying sizes and composition (DNA, extent of cell wall, etc.). The authors have termed these vesicles 'S-form cells', and distinguish them from the similar-appearing L-forms in that they originate as a result of exposure to osmotic stress, as opposed to in response to cell wall-targeting enzymes/antibiotics. These S-form cells appear to be able to replicate in a manner reminiscent to that reported previously for L-forms, contain DNA but appears to lack a conventional segregation mechanism as many lose a resident megaplasmid, and may be able to switch back to conventional mycelial growth.

1- While the S-form cells shown here are striking, it isn't clear how these differ from L-forms, which have previously been reported for the *K. viridifaciens* strain investigated here (previously known as *S. viridifaciens* – Innes et al 2001). Based on a recent review by Errington and colleagues (2016), L-form cells are loosely defined as: "variants of normally walled bacteria that have adapted to grow in the complete absence of cell wall synthesis". While the S-form cells seem to be generally larger than L-form ones, there is no evidence presented that suggests other obvious differences. Also the consistently large S-form cells seem to be specific to *K. viridifaciens*, based on the data presented in Extended Data Table 4.

2- Additional membrane synthesis would make sense in order to facilitate the release of these membrane vesicle-like cells. However, the information provided in Extended Data Tables 1 and 2 is challenging to understand. What is meant by 'branching frequency'? What does 'membrane fraction' mean? – would it not be expected that all hyphae should be encased by membranes? I think these are explained in the methods, but it would also be useful to have this info in the Table itself. How is it decided where a hypha starts and ends with respect to defining membrane fraction (and presumably 'length')? Collecting data for only 10 hypha seems low.

3- It isn't entirely clear from the methods description, how confident one could be about the lack

of contaminating spores or small mycelia that may have passed through the filtration steps used to collect S-cells (lines 406-421). It sounds like the lysozyme treatment (what is in the 'lysozyme solution? – please provide a recipe or reference) used to remove residual hyphal fragments may actually help in promoting L-forms, and it is not clear how such a treatment would remove hyphal fragments, nor how it would impact any contaminating spores apart from possibly helping to promote germination. Are there any fluorescent markers – e.g. specific for spores – that could be included as an additional internal check to ensure no spore contamination? I have watched the S4 Video repeatedly, and at least some of the mycelial outgrowth seems to be coming from small spore-like cells (in other cases I can't tell where they originate).

4- The frequent loss of the megaplasmid KVP1 after re-growth of S-cells on MYM medium is interesting. Is a similar thing seen if osmotically stressed cells (not filtered for S-cells) are plated? It would be interesting to know if this was a general response to osmotic stress, or if this was something specific to S-cells. This could be tested using strains that aren't known to form S-cells (e.g. *S. coelicolor*).

5- The M1/M2 mutants are particularly fascinating, and additional follow up work on these strains would help to further strengthen the manuscript. It would be important to complement the mutations associated with these two strains, as that would provide important insight into the basis for S-form proliferation. If it isn't possible to complement the mutant phenotype, then it may suggest a role for gene regulation or epigenetic factors.

When considering the M1 and M2 strains, were these strains viable when plated without high osmolarity? If yes, what did the colonies look like? Were they able to return to mycelial-type growth, or did the cells all die?

6- The idea put forward in the discussion, suggesting that S-cells may be able to take up DNA from the environment is intriguing but at this point is entirely speculative. In the absence of additional experimental support, it would be worth toning down this section. It seems like an idea that would be very straightforward to test in principle: add DNA with an antibiotic resistance gene on it and see if it gets taken up/incorporated into the chromosome by testing for resistance.

7- Do the authors have any thoughts as to why some strains appeared to generate S-cells whereas others did not? This would seem like an appropriate discussion point.

8- Figure comments:

Figure S1D- at 96h, there seems to be little overlap between the membrane and DNA stains, and no obvious S-cells. Can the authors comment?

Figure S2 (and throughout the manuscript)- it is not clear why the authors indicate that WGA stains newly synthesised peptidoglycan. Could it not also stain old PG? Also, what concentration of sucrose are these cells grown in?

Figure S3- S-cells are not readily visible for many of the MBT strains in the bright-field images provided. Was an S-cell designation made/guided by DNA and membrane stains instead?

Figure S5- it is not clear what this image is showing

Reviewer #3 (Remarks to the Author):

Although cell wall is normally essential structure for the bacterial growth and viability, it has been

known that under certain conditions many bacteria are able to switch into a cell wall-deficient (CWD) state, which is completely resistant to many antibiotics working on cell wall synthesis. This is an interesting study on still poorly understood but highly interesting cellular pathways involved in the generation of CWD bacteria. The switch to CWD state from parental walled cells is generally induced by directly inhibiting cell wall synthesis with antibiotics, lysozyme and/or genetic modifications under osmoprotective conditions. In this work the authors convincingly show that, in a wide range of filamentous actinobacteria, the CWD state (S-cells) can be induced in the presence of high levels of sucrose or NaCl, even without artificial direct inhibition of cell wall synthesis. The authors also show that the S-cells can further develop to S-forms, which can proliferate without walls, by prolonged culture in the presence of sucrose or NaCl. It is not clear why sucrose or NaCl should promote the switch to S-forms, but this mechanism is probably beyond the scope of this paper. The manuscript is very well written and the experiments are sound. This study would attract a broad audience interested in basic bacterial physiology responding to environmental effects.

Specific comments

The key initial observation in this work is the inhibition of tip growth and the generation of excess membrane in the presence of higher levels of osmolytes (sucrose or NaCl) in walled cells. The membrane-bound S-cells are released from hyphal tips on the original cell wall probably due to ongoing autolytic activity, suggesting uncoupling of synthesis of the cell envelope layers is induced by the osmolytes. When the S-cells are cultured for prolonged periods in the presence of osmolytes, they are able to switch to S-forms. The authors conclude that the S-form switch is induced by hyperosmotic stress. However, on the other hand, the osmolytes also work to provide an isotonic environment, sufficiently osmoprotective to support viability/growth of these CWD bacteria. In this context, it seems to be not really clear about significance of hyperosmotic stress on generation of CWD bacteria, especially for S-forms. Excess sucrose or NaCl could also bring toxic side effects, other than hyperosmotic stress.

- 1) Sucrose or its derivatives may be taken and utilised in actinomycetes? Utilization of excess amount of sugar might affect growth and viability.
- 2) NaCl causes not only osmotic stress but also ionic toxicity.

The possible effects of sucrose or NaCl on the generation or selection of CWD bacteria should be more carefully interpreted.

Rebuttal – Stress-induced formation of cell wall-deficient cells in filamentous actinomycetes

Reviewers' comments:

Reviewer #1 (Remarks to the Author):

This a fascinating study and although I have no issues with the main findings, there are some aspects of the manuscript that require attention.

>>> We thank the referee for the positive remarks

Abstract:

It is an over-interpretation to state that this study provides important insights into how genes may disseminate in the environment. This has not been addressed experimentally and is pure speculation. I recommend removing it from the Abstract, but it is worthwhile mentioning in the Discussion.

>>> We have removed this speculation in the abstract of the revised manuscript

Results:

Loss of KVP1 - the developmental phenotypes described are very reminiscent of the results of genetic instability in *Streptomyces*, due to loss of large amounts of the terminal regions of the chromosome. Loss of plasmid could be verified by PFGE and, at the same time, this would permit analysis of the chromosome. The results of genome sequencing could also confirm the nature of the genetic changes, although only results of a few SNPs are reported. Does the sequence analysis confirm the complete absence of the megaplasmid? Is there evidence for loss of any other genes?

>>> We agree that the developmental phenotypes observed after S-cell switching are reminiscent of the phenotypes associated with genetic instability in *Streptomyces*, which we now mentioned in the discussion of the revised manuscript (see p. 11, lines 331-333). Unfortunately, we have no genome sequence information of the R3-R5 strains (the SNPs the referee refers to were derived from genome sequences of the M1, M2 and *alpha* mutant strains, see below). In a recently started new PhD project, however, we are analyzing the instability phenotypes and the biological consequences of this heterogeneity in more detail, and which will be published separately.

We are confident that the KVP1 megaplasmid is no longer present in the R3-R5 mutants (see Fig. 4). In particular, we wish to emphasize that we tested for the presence of the KVP1-located *parA* and *orf1* genes, which are required for plasmid segregation and replication, respectively^{1,2}. We have added a sentence in the manuscript to emphasize this, see p. 8, lines 222-223).

While we currently have no additional genome sequence information of the R3-R5 strains, we have performed additional analyses using the mutants M1, M2 and *alpha* that are able to proliferate without the cell wall, and which also appear to have lost the KVP1 megaplasmid. Notably, alignment of the sequencing data to the genome sequence of *Kitasatospora viridifaciens* not only confirms loss of the megaplasmid-encoded genes, but also infers changes in the chromosome (see Fig. S7). In particular, the sequences at the right arm of the chromosome appear to be less abundantly present compared to the rest of the chromosome. We have included these results in the revised version of the manuscript (see p. 10, lines 279-281).

Reviewer #2 (Remarks to the Author):

The work of Ramijan and colleagues describes the intriguing observation that some sporulating actinobacteria grown under high osmotic conditions extrude membrane vesicles of varying sizes and composition (DNA, extent of cell wall, etc.). The authors have termed these vesicles 'S-form cells', and distinguish them from the similar-appearing L-forms in that

they originate as a result of exposure to osmotic stress, as opposed to in response to cell wall-targeting enzymes/antibiotics. These S-form cells appear to be able to replicate in a manner reminiscent to that reported previously for L-forms, contain DNA but appears to lack a conventional segregation mechanism as many lose a resident megaplasmid, and may be able to switch back to conventional mycelial growth.

1- While the S-form cells shown here are striking, it isn't clear how these differ from L-forms, which have previously been reported for the *K. viridifaciens* strain investigated here (previously known as *S. viridifaciens* – Innes et al 2001). Based on a recent review by Errington and colleagues (2016), L-form cells are loosely defined as: "variants of normally walled bacteria that have adapted to grow in the complete absence of cell wall synthesis". While the S-form cells seem to be generally larger than L-form ones, there is no evidence presented that suggests other obvious differences. Also the consistently large S-form cells seem to be specific to *K. viridifaciens*, based on the data presented in Extended Data Table 4.

>>> We thank the referee for the constructive feedback. We noticed that the terms used for the various cell wall-deficient cells (S-cells, S-forms and L-forms) in the original manuscript were confusing. S-cells are the cell-wall deficient cells extruded by filaments in a range of actinomycetes, and which are not able to proliferate without the cell wall. Instead S-cells will revert back to the mycelial mode-of-growth. By contrast, S-forms (strains M1 and M2) and L-forms are able to proliferate in the cell wall-deficient state. In the revised manuscript, we have renamed S-forms to L-forms, also because they are morphologically indistinguishable and behave identical (see also the additional Fig. 5A/5B). The only difference between them is their origin: L-forms are derived from chemical and/or enzymatic induction, while the M1/M2 strains described in this paper originate solely from prolonged exposure to hyperosmotic stress.

2- Additional membrane synthesis would make sense in order to facilitate the release of these membrane vesicle-like cells. However, the information provided in Extended Data Tables 1 and 2 is challenging to understand. What is meant by 'branching frequency'? What does 'membrane fraction' mean? – would it not be expected that all hyphae should be encased by membranes? I think these are explained in the methods, but it would also be useful to have this info in the Table itself. How is it decided where a hypha starts and ends with respect to defining membrane fraction (and presumably 'length')? Collecting data for only 10 hypha seems low.

>>> We have included a new supplementary figure (Fig. S1) that details how the quantification was performed and how the different parameters indicated in Extended Data Tables 1 and 2 were obtained. We also added more textual information on the methodology in the Table legends (see p. 30/31 of the revised manuscript). We feel that quantifying 10 hyphae from 10 independent pellets, which all need to be analyzed manually, is sufficient for two reasons: first, the differences between hyphae exposed to and those not exposed to hyperosmotic stress is majorly different. And secondly, our results are in full agreement with data presented in Fuchino *et al.*, 2017, where morphological changes accompanying osmotic shocks in *Streptomyces coelicolor* were analysed³.

3- It isn't entirely clear from the methods description, how confident one could be about the lack of contaminating spores or small mycelia that may have passed through the filtration steps used to collect S-cells (lines 406-421). It sounds like the lysozyme treatment (what is in the 'lysozyme solution'? – please provide a recipe or reference) used to remove residual hyphal fragments may actually help in promoting L-forms, and it is not clear how such a treatment would remove hyphal fragments, nor how it would impact any contaminating spores apart from possibly helping to promote germination. Are there any fluorescent markers – e.g. specific for spores – that could be included as an additional internal check to ensure no spore contamination? I have watched the S4 Video repeatedly, and at least some

of the mycelial outgrowth seems to be coming from small spore-like cells (in other cases I can't tell where they originate).

>>> We have constructed an *ssgB* mutant in *Kitasatospora viridifaciens* that is no longer able to sporulate⁴ (see the new Fig. 3H). Isolated S-cells from this mutant strain (Fig. 3I) were also able to revert to the mycelial mode-of-growth. We replaced the original movie of the WT strain with one of the *ssgB* mutant, which we believe more convincingly shows the switch to the mycelial mode-of-growth (Fig. 3J). Switching is typically not very efficient for reasons that are currently unknown, which is something that has also been reported for other cell wall-deficient cells, including protoplasts and L-forms^{5,6}. This makes capturing these events very difficult to trap. Please note that we performed the experiments with the *ssgB* mutant S-cells without using lysozyme.

4- The frequent loss of the megaplasmid KVP1 after re-growth of S-cells on MYM medium is interesting. Is a similar thing seen if osmotically stressed cells (not filtered for S-cells) are plated? It would be interesting to know if this was a general response to osmotic stress, or if this was something specific to S-cells. This could be tested using strains that aren't known to form S-cells (e.g. *S. coelicolor*).

>>> Large megaplasmids, which are not uncommon in filamentous actinomycetes, have typically a low copy number^{2,7}. Plasmid loss has also been reported following protoplast regeneration in *Streptomyces clavuligerus*². In agreement, loss of the KVP1 megaplasmid is also easily observed when protoplasts of *K. viridifaciens* are plated, but rare when spores are used, which we now added as new data to the revised manuscript (see Figs. 4A-C). We expect that during their formation, a fraction of the protoplasts and S-cells will simply not possess the megaplasmid. This may be caused by substantial differences in the abundance and localization of the megaplasmid in certain sections of the hyphae. We have added this information in a new paragraph in the Discussion (see p. 11, lines 330-342 of the revised manuscript).

5- The M1/M2 mutants are particularly fascinating, and additional follow up work on these strains would help to further strengthen the manuscript. It would be important to complement the mutations associated with these two strains, as that would provide important insight into the basis for S-form proliferation. If it isn't possible to complement the mutant phenotype, then it may suggest a role for gene regulation or epigenetic factors.

>>> We agree with the referee that further characterization of the mutations is interesting. Importantly though, we noticed that in addition to the reported SNPs, other more profound changes were evident in the chromosomes of the M1 and M2 mutants (see the additional Fig. S7D-E). As such, we cannot rule out that those changes are causing the M1 and M2 strains to proliferate. We are currently analyzing these results in more detail, which we expect to publish separately in the future. The main point we wanted to make here is that the proliferating cell wall-deficient variants (i.e. M1 and M2) are mutants.

When considering the M1 and M2 strains, were these strains viable when plated without high osmolarity? If yes, what did the colonies look like? Were they able to return to mycelial-type growth, or did the cells all die?

>>> We performed these experiments and can confirm that those strains are viable (and walled) when plated on media without high osmolarity. These results have been added in the revised manuscript (see p. 9, lines 251-255 and Fig. 5B).

6- The idea put forward in the discussion, suggesting that S-cells may be able to take up DNA from the environment is intriguing but at this point is entirely speculative. In the absence of additional experimental support, it would be worth toning down this section. It seems like an idea that would be very straightforward to test in principle: add DNA with an antibiotic resistance gene on it and see if it gets taken up/incorporated into the chromosome by testing for resistance.

>>> We have toned down this section in the discussion and removed the speculation entirely from the abstract (see also comment Referee 1).

7- Do the authors have any thoughts as to why some strains appeared to generate S-cells whereas others did not? This would seem like an appropriate discussion point.

>>> We have a stringent method to assess S-cell formation, which we define as cell wall-deficient cells that contain DNA and which are larger in size than 2 μm (to exclude that these are swollen spores). In this study, strains were screened in liquid-grown cultures while shaking at 100 RPM (see Materials and Methods). Being this stringent, we think that potentially many more strains are able to make S-cells, but not under the conditions that were tested here. We have added this in the Discussion of the revised manuscript (see p. 11, lines 320-329).

8- Figure comments:

Figure S1D- at 96h, there seems to be little overlap between the membrane and DNA stains, and no obvious S-cells. Can the authors comment?

>>> Growth in NaCl leads to pelleted growth, i.e. dense particles are formed, which not grow very efficiently and complicate imaging and visualization of individual hyphae. We agree that there was little overlap visible in the original panels. We have now taken a new set of images (see Fig. S3), where the overlap between membrane and DNA is more evident. This panel also shows S-cells.

Figure S2 (and throughout the manuscript)- it is not clear why the authors indicate that WGA stains newly synthesised peptidoglycan. Could it not also stain old PG? Also, what concentration of sucrose are these cells grown in?

>>> WGA binds to areas of the wall that are either not yet fully polymerized or are being autolyzed^{8,9}. We have removed the word nascent in the text of the revised manuscript.

The concentration of sucrose used was 0.6 M, which is now added in the legend of Fig. S4 of the revised version.

Figure S3- S-cells are not readily visible for many of the MBT strains in the bright-field images provided. Was an S-cell designation made/guided by DNA and membrane stains instead?

>>> We have replaced some panels in Fig. S3 (now Fig. S5) to better visualize S-cells. The panels that were replaced are those for strains MBT13, MBT63 and MBT64. As discussed in point 7, we define S-cells as vesicles (visible with BF) that contain DNA and are larger than 2 μm (to exclude that these could represent swollen spores).

Figure S5- it is not clear what this image is showing

>>> We agree with the referee that this image is not clear, and we have decided to remove it.

Reviewer #3 (Remarks to the Author):

Although cell wall is normally essential structure for the bacterial growth and viability, it has been known that under certain conditions many bacteria are able to switch into a cell wall-deficient (CWD) state, which is completely resistant to many antibiotics working on cell wall synthesis. This is an interesting study on still poorly understood but highly interesting cellular pathways involved in the generation of CWD bacteria. The switch to CWD state from parental walled cells is generally induced by directly inhibiting cell wall synthesis with antibiotics, lysozyme and/or genetic modifications under osmoprotective conditions. In this work the authors convincingly show that, in a wide range of filamentous actinobacteria, the CWD state (S-cells) can be induced in the presence of high levels of sucrose or NaCl, even without artificial direct inhibition of cell wall synthesis. The authors also show that the S-cells can further develop to S-forms, which can proliferate without walls, by prolonged culture in

the presence of sucrose or NaCl. It is not clear why sucrose or NaCl should promote the switch to S-forms, but this mechanism is probably beyond the scope of this paper. The manuscript is very well written and the experiments are sound. This study would attract a broad audience interested in basic bacterial physiology responding to environmental effects. >>> We thank the referee for the positive remarks. Indeed, the mechanism behind the conversion of S-cells to proliferating S-forms (now renamed to L-forms in the revised version, see comments Referee 2) is beyond the scope of this paper.

Specific comments

The key initial observation in this work is the inhibition of tip growth and the generation of excess membrane in the presence of higher levels of osmolytes (sucrose or NaCl) in walled cells. The membrane-bound S-cells are released from hyphal tips on the original cell wall probably due to ongoing autolytic activity, suggesting uncoupling of synthesis of the cell envelope layers is induced by the osmolytes. When the S-cells are cultured for prolonged periods in the presence of osmolytes, they are able to switch to S-forms. The authors conclude that the S-form switch is induced by hyperosmotic stress. However, on the other hand, the osmolytes also work to provide an isotonic environment, sufficiently osmoprotective to support viability/growth of these CWD bacteria. In this context, it seems to be not really clear about significance of hyperosmotic stress on generation of CWD bacteria, especially for S-forms. Excess sucrose or NaCl could also bring toxic side effects, other than hyperosmotic stress.

1) Sucrose or its derivatives may be taken and utilised in actinomycetes? Utilization of excess amount of sugar might affect growth and viability.

2) NaCl causes not only osmotic stress but also ionic toxicity.

The possible effects of sucrose or NaCl on the generation or selection of CWD bacteria should be more carefully interpreted.

>>> This is an interesting point that the referee considers. Indeed, on the one hand we use these compounds to generate hyperosmotic stress, while on the other hand they are also required for survival of the CWD cells. Their protective role relates to the high turgor pressure in the cytosol, which generates an outward pressure on the cell membrane. To prevent catastrophic cell lysis without a cell wall, high levels of solutes are needed to counterbalance the internal turgor pressure. In a recent study high levels of sucrose were used to study the osmotic stress response of *Streptomyces coelicolor*³. In that study, mycelium was exposed to 1 M sucrose, twice the amount used in our study, in so-called YEME medium, which is commonly used in the *Streptomyces* community¹⁰. Sucrose may be utilized by some actinomycetes, but this certainly is very rare¹⁰.

While we cannot exclude any toxic side-effects of NaCl and potential consumption of sucrose at this moment, we want to emphasize that these osmolytes are very different from one another, and are generally accepted and used as compounds to study the effect of (hyper)osmotic stress¹¹⁻¹³. Notably, we did observe a difference in morphology when mycelium was exposed to NaCl as compared to sucrose. In particular, the mycelial particles that were formed were more dense and smaller than those obtained in media containing high levels of sucrose. We have mentioned this in the revised version of the manuscript (see p. 5, lines 123-125).

To further substantiate our findings, we also tested whether S-cells were formed in the presence of sorbitol. This alcohol is also commonly used as an osmolyte¹³. When cells were grown in the presence of 1 M sorbitol, we also noticed a dramatic reduction in growth coinciding with the appearance of S-cells in the culture (see Fig. XXX in the revised manuscript). Altogether, we feel that sufficiently warrant the conclusion that a severe reduction in growth, due to hyperosmotic stress, leads to the formation of S-cells. Prolonged exposure to such hyperosmotic conditions can then lead to mutations and the formation of S-forms (which are now called L-forms in the revised manuscript, see comments Referee 2).

References

- 1 Medema, M. H. *et al.* The sequence of a 1.8-mb bacterial linear plasmid reveals a rich evolutionary reservoir of secondary metabolic pathways. *Genome Biol Evol* **2**, 212-224, doi:10.1093/gbe/evq013 (2010).
- 2 Álvarez-Álvarez, R. *et al.* A 1.8-Mb-reduced *Streptomyces clavuligerus* genome: relevance for secondary metabolism and differentiation. *Appl Microbiol Biotechnol* **98**, 2183-2195, doi:10.1007/s00253-013-5382-z (2014).
- 3 Fuchino, K., Flärdh, K., Dyson, P. & Ausmees, N. Cell-biological studies of osmotic shock response in *Streptomyces* spp. *J Bacteriol* **199**, doi:10.1128/JB.00465-16 (2017).
- 4 Willemse, J., Borst, J. W., de Waal, E., Bisseling, T. & van Wezel, G. P. Positive control of cell division: FtsZ is recruited by SsgB during sporulation of *Streptomyces*. *Genes Dev* **25**, 89-99, doi:10.1101/gad.600211 (2011).
- 5 Bourne, N. & Dancer, B. N. Regeneration of protoplasts of *Bacillus subtilis* 168 and closely related strains. *J Gen Microbiol* **132**, 251-255, doi:10.1099/00221287-132-2-251 (1986).
- 6 Mercier, R., Kawai, Y. & Errington, J. Excess membrane synthesis drives a primitive mode of cell proliferation. *Cell* **152**, 997-1007, doi:10.1016/j.cell.2013.01.043 (2013).
- 7 Kinashi, H. & Shimaji-Murayama, M. Physical characterization of SCP1, a giant linear plasmid from *Streptomyces coelicolor*. *J Bacteriol* **173**, 1523-1529 (1991).
- 8 Schwedock, J., McCormick, J. R., Angert, E. R., Nodwell, J. R. & Losick, R. Assembly of the cell division protein FtsZ into ladder-like structures in the aerial hyphae of *Streptomyces coelicolor*. *Mol Microbiol* **25**, 847-858 (1997).
- 9 Allen, A. K., Neuberger, A. & Sharon, N. The purification, composition and specificity of wheat-germ agglutinin. *Biochem J* **131**, 155-162 (1973).
- 10 Kieser, T., Bibb, M. J., Buttner, M. J., Chater, K. F. & Hopwood, D. A. *Practical Streptomyces genetics*. (The John Innes Foundation, 2000).
- 11 López, C. S. *et al.* Biochemical and biophysical studies of *Bacillus subtilis* envelopes under hyperosmotic stress. *Int J Food Microbiol* **55**, 137-142 (2000).
- 12 Dai, X. *et al.* Slowdown of translation elongation in *Escherichia coli* under hyperosmotic stress. *mBio* **9**, e02375-02317 (2018).
- 13 Miermont, A. *et al.* Severe osmotic compression triggers a slowdown of intracellular signaling, which can be explained by molecular crowding. *Proc Natl Acad Sci U S A* **110**, 5725-5730, doi:10.1073/pnas.1215367110 (2013).

REVIEWERS' COMMENTS:

Reviewer #1 (Remarks to the Author):

The authors have addressed the major points raised in review. I am happy with the revised manuscript.

Reviewer #2 (Remarks to the Author):

In the revised manuscript of Ramijan et al., the authors have very nicely addressed the reviewers' comments from the initial round of review. Consequently the majority of my comments are relatively minor.

Lines 90-91- Confusing – please rephrase. It is not clear how 'so-called L-forms' differ from 'L-forms'.

Line 93- While S-cells and L-forms in filamentous actinobacteria do seem to arise during prolonged exposure to osmotic cells, it remains possible that other conditions may promote such cell formation. It would be worth removing the 'solely' descriptor.

Lines 99-100- While high concentrations of sucrose had an adverse effect on growth, lower levels appeared to improve growth. It would be worth rephrasing the 'increasing amounts of sucrose' statement to instead read something like 'high levels of sucrose'.

Lines 104 and 106- How reproducible were these observations? How many colonies were measured? How many biological replicates were examined for the CFU counts?

Lines 135- It would be worth indicating how many cells were counted (>200/cell type)

Lines 306-7- While this work provides compelling evidence that a subset of filamentous actinobacteria can release S-cells and develop L-forms, this does not necessarily constitute evidence that such cells have ecological relevance. Please tone down this statement.

Line 453: grown?

Fig S1- 'length' is misspelled

Fig S2 & S3- it would be nice to have a low osmolarity control sample shown – this would allow readers to better appreciate the dense pellets and the weird vesicles

Reviewer #3 (Remarks to the Author):

The authors responded well to my original comments. I support publication in Nature communications and think that would reach a broad audience of readers.

REVIEWERS' COMMENTS:

Reviewer #1 (Remarks to the Author):

The authors have addressed the major points raised in review. I am happy with the revised manuscript.

>>> We would like to thank Reviewer 1 for the constructive contribution during the review process.

Reviewer #2 (Remarks to the Author):

In the revised manuscript of Ramijan et al., the authors have very nicely addressed the reviewers' comments from the initial round of review. Consequently the majority of my comments are relatively minor.

>>> We thank Reviewer 2 for the positive words and constructive feedback.

Lines 90-91- Confusing – please rephrase. It is not clear how 'so-called L-forms' differ from 'L-forms'.

>>> We have rephrased the sentence.

Line 93- While S-cells and L-forms in filamentous actinobacteria do seem to arise during prolonged exposure to osmotic cells, it remains possible that other conditions may promote such cell formation. It would be worth removing the 'solely' descriptor.

>>> Changed as suggested.

Lines 99-100- While high concentrations of sucrose had an adverse effect on growth, lower levels appeared to improve growth. It would be worth rephrasing the 'increasing amounts of sucrose' statement to instead read something like 'high levels of sucrose'.

>>> Changed as suggested.

Lines 104 and 106- How reproducible were these observations? How many colonies were measured? How many biological replicates were examined for the CFU counts?

>>> We have added more information on the reproducibility and CFU numbers in the text and in the figure legend (see lines 94-97 of the revised manuscript).

Lines 135- It would be worth indicating how many cells were counted (>200/cell type)

>>> The number of counted cells has been added to the revised version of the manuscript (see lines 125-126 of the revised manuscript).

Lines 306-7- While this work provides compelling evidence that a subset of filamentous actinobacteria can release S-cells and develop L-forms, this does not necessarily constitute evidence that such cells have ecological relevance. Please tone down this statement.

>>> We have toned down this statement and rephrased the sentence into: "This work thus provides leads towards dissecting the ecological relevance that such cells may have."

Line 453: grown?

>>> Changed as suggested.

Fig S1- 'length' is misspelled

>>> Changed as suggested.

Fig S2 & S3- it would be nice to have a low osmolarity control sample shown – this would allow readers to better appreciate the dense pellets and the weird vesicles

>>> We have added control panels as suggested.

Reviewer #3 (Remarks to the Author):

The authors responded well to my original comments. I support publication in Nature communications and think that would reach a broad audience of readers.

>>> We would like to thank Reviewer 3 for the constructive contribution during the review process.